# New fossils of *Australopithecus sediba* reveal a nearly complete lower back

Scott A Williams[1,2,3,4]*, Thomas Cody Prang[5], Marc R Meyer[6], Thierra K Nalley[7], Renier Van Der Merwe[3], Christopher Yelverton[4,8], Daniel García-Martínez[3,9,10], Gabrielle A Russo[11], Kelly R Ostrofsky[12], Jeffrey Spear[1,2], Jennifer Eyre[1,13], Mark Grabowski[14], Shahed Nalla[3,15], Markus Bastir[3,16], Peter Schmid[3,17], Steven E Churchill[3,18], Lee R Berger[3]

[1]Center for the Study of Human Origins, Department of Anthropology, New York University, New York, United States; [2]New York Consortium in Evolutionary Primatology, New York, United States; [3]Centre for the Exploration of the Deep Human Journey, University of the Witwatersrand, Johannesburg, South Africa; [4]Evolutionary Studies Institute, University of the Witwatersrand, Johannesburg, South Africa; [5]Department of Anthropology, Texas A&M University, College Station, United States; [6]Department of Anthropology, Chaffey College, Rancho Cucamonga, United States; [7]Western University of Health Sciences, College of Osteopathic Medicine of the Pacific, Department of Medical Anatomical Sciences, Pomona, United States; [8]Department of Chiropractic, Faculty of Health Sciences, University of Johannesburg, Johannesburg, South Africa; [9]Centro Nacional de Investigación sobre la Evolución Humana (CENIEH), Burgos, Spain; [10]Departamento de Biodiversidad, Ecología y Evolución, Universidad Complutense de Madrid (UCM), Madrid, Spain; [11]Department of Anthropology, Stony Brook University, Stony Brook, United States; [12]Department of Anatomy, College of Osteopathic Medicine, New York Institute of Technology, Old Westbury, United States; [13]Department of Anthropology, Bryn Mawr College, Bryn Mawr, United States; [14]Research Centre in Evolutionary Anthropology and Palaeoecology, Liverpool John Moores University, Liverpool, United Kingdom; [15]Department of Human Anatomy and Physiology, Faculty of Health Sciences, University of Johannesburg, Johannesburg, South Africa; [16]Departamento de Paleobiología, Museo Nacional de Ciencias Naturales (CSIC), Madrid, Spain; [17]Anthropological Institute and Museum, University of Zurich, Zurich, Switzerland; [18]Department of Evolutionary Anthropology, Duke University, Durham, United States

*For correspondence: sawilliams@nyu.edu

Competing interest: The authors declare that no competing interests exist.

**Abstract** Adaptations of the lower back to bipedalism are frequently discussed but infrequently demonstrated in early fossil hominins. Newly discovered lumbar vertebrae contribute to a near-complete lower back of Malapa Hominin 2 (MH2), offering additional insights into posture and locomotion in *Australopithecus sediba*. We show that MH2 possessed a lower back consistent with lumbar lordosis and other adaptations to bipedalism, including an increase in the width of inter-vertebral articular facets from the upper to lower lumbar column ('pyramidal configuration'). These results contrast with some recent work on lordosis in fossil hominins, where MH2 was argued to demonstrate no appreciable lordosis ('hypolordosis') similar to Neandertals. Our three-dimensional geometric morphometric (3D GM) analyses show that MH2's nearly complete middle lumbar vertebra is human-like in overall shape but its vertebral body is somewhat intermediate in shape between modern humans and great apes. Additionally, it bears long, cranially and ventrally oriented costal (transverse) processes, implying powerful trunk musculature. We interpret this combination

of features to indicate that *A. sediba* used its lower back in both bipedal and arboreal positional behaviors, as previously suggested based on multiple lines of evidence from other parts of the skeleton and reconstructed paleobiology of *A. sediba*.

## Introduction

Bipedal locomotion is thought to be one of the earliest and most extensive adaptations in the hominin lineage, potentially evolving initially 6–7 million years (Ma) ago. Human-like bipedalism evolved gradually, however, and early hominins appear to have been facultative bipeds on the ground and competent climbers in the trees (*Senut et al., 2001*; *White et al., 2015*; *Prang, 2019*; *Prang et al., 2021*). How long climbing adaptations persisted in hominins and when adaptations to obligate terrestrial bipedalism evolved are major outstanding questions in paleoanthropology. *Australopithecus sediba* – an early Pleistocene (~2 Ma) hominin from the site of Malapa, Gauteng province, South Africa – has featured prominently in these discussions, as well as those concerning the origins of the genus *Homo* (*Berger et al., 2010*; *Berger, 2012*; *Irish et al., 2013*; *Dembo et al., 2015*; *Kimbel and Rak, 2017*; *De Ruiter et al., 2018*; *Williams et al., 2018a*; *Du and Alemseged, 2019*).

Previous studies support the hypothesis that *A. sediba* possessed adaptations to arboreal locomotion and lacked traits reflecting a form of obligate terrestriality observed in later hominins (*Schmid et al., 2013*; *Prang, 2015a*; *Prang, 2015b*; *Prang, 2016*; *Holliday et al., 2018*). Malapa Hominin 2 (MH2) metacarpals are characterized by trabecular density most similar to orangutans, which suggests power grasping capabilities (*Dunmore et al., 2020*), and the MH2 ulna was estimated to reflect a high proportion of forelimb suspension in the locomotor repertoire of *A. sediba* (*Rein et al., 2017*). Evidence from the lower limb also suggests that *A. sediba* lacked a robust calcaneal tuber (*Prang, 2015a*) and a longitudinal arch (*Prang, 2015b*), both thought to be adaptations to obligate, human-like bipedalism, and demonstrates evidence for a mobile subtalar joint proposed to be adaptively significant for vertical climbing and other arboreal locomotor behaviors (*Prang, 2016*; *DeSilva et al., 2013*; *Zipfel et al., 2011*; *DeSilva et al., 2018*). The upper thorax (*Schmid et al., 2013*), scapula (*Churchill et al., 2013*; *Churchill et al., 2018*), and cervical vertebrae (*Meyer et al., 2017*) of *A. sediba* suggest shoulder and arm elevation indicative of arboreal positional behaviors requiring overhead arm positions, and the limb joint size proportions are ape-like (*Prabhat et al., 2021*). Furthermore, analysis of dental calculus from Malapa Hominin 1 (MH1) indicates that this individual's diet was high in $C_3$ plants like fruit and leaves, similar to savannah chimpanzees and *Ardipithecus ramidus* (*Henry et al., 2012*).

Despite the presence of climbing adaptations, *A. sediba* also demonstrates clear evidence for bipedal locomotion. The knee and ankle possess human-like adaptations to bipedalism, demonstrating a valgus angle of the femur and a human-like ankle joint (*Zipfel et al., 2011*; *DeSilva et al., 2013*; *DeSilva et al., 2018*). Evidence for strong dorsal (lordotic) wedging of the two lower lumbar vertebrae suggests the presence of a lordotic (ventrally convex) lower back (*Williams et al., 2013*; *Williams et al., 2018b*). However, the initial recovery of just the last two lumbar vertebrae of MH2 limited interpretations of spinal curvature, and a study of the MH2 pelvis reconstruction (*Kibii et al., 2011*) suggests that *A. sediba* was characterized by a small lordosis angle estimated from calculated pelvic incidence (*Been et al., 2014*). A separate pelvis reconstruction of MH2 produces a pelvic incidence angle more in line with other hominins (*Tardieu et al., 2017*). The presence of a long, mobile lower back and a *Homo*-like lower thorax morphology indicating the presence of a waist further suggest bipedal adaptations in *A. sediba* (*Schmid et al., 2013*; *Williams et al., 2013*). However, missing and incomplete lumbar vertebrae prevented comparative analysis of overall lower back morphology and allowed only limited interpretations of *A. sediba* back posture and implications for positional behavior.

Here, we report the discovery of portions of four lumbar vertebrae from two ex situ breccia blocks that were excavated from an early 20th century mining road and dump at Malapa. The former mining road is represented by a trackway located in the northern section of the site approximately 2 m north of the main pit that yielded the original *A. sediba* finds (*Dirks et al., 2010*; *Figure 1*). The trackway traverses the site in an east-west direction and was constructed using breccia and soil removed from the main pit by the historic limestone miners. Specimens U.W.88–232, −233,−234, and −281 were recovered in 2015 from the upper section of layer 2 (at a depth of 10 cm) and formed part of the foundation layer of the mining road. The trackway can be distinguished from the surrounding deposits

**eLife digest** One of the defining features of humans is our ability to walk comfortably on two legs. To achieve this, our skeletons have evolved certain physical characteristics. For example, the lower part of the human spine has a forward curve that supports an upright posture; whereas the lower backs of chimpanzees and other apes – which walk around on four limbs and spend much of their time in trees – lack this curvature. Studying the fossilized back bones of ancient human remains can help us to understand how we evolved these features, and whether our ancestors moved in a similar way.

*Australopithecus sediba* was a close-relative of modern humans that lived about two million years ago. In 2008, fossils from an adult female were discovered at a cave site in South Africa called Malapa. However, the fossils of the lower back region were incomplete, so it was unclear whether the female – referred to as Malapa Hominin 2 (MH2) – had a forward-curving spine and other adaptations needed to walk on two legs.

Here, Williams et al. report the discovery of new *A. sediba* fossils from Malapa. The new fossils are mainly bones from the lower back, and they fit together with the previously discovered MH2 fossils, providing a nearly complete lower spine. Analysis of the fossils suggested that MH2 would have had an upright posture and comfortably walked on two legs, and the curvature of their lower back was similar to modern females. However, other aspects of the bones' shape suggest that as well as walking, *A. sediba* probably spent a significant amount of time climbing in trees.

The findings of Williams et al. provide new insights in to our evolutionary history, and ultimately, our place in the natural world around us. Our lower back is prone to injury and pain associated with posture, pregnancy and exercise (or lack thereof). Therefore, understanding how the lower back evolved may help us to learn how to prevent injuries and maintain a healthy back.

by a section of compacted soil (comprising quartz, cherts, and flowstone) and breccia that extends between layers 1 and 2. Breccia recovered from the trackway, including the block containing U.W.88–232, −233,–234, and −281 similarly presented with quantities of embedded quartz fragments and grains. The breccia block containing specimen U.W.88–280, along with U.W.88–43, –44, and –114 (*Williams et al., 2013*; *Williams et al., 2018b*), were recovered from the miner's dump comprised of excess material (soil and breccia) used for the construction of the miner's road. The composition of the road matrix and associated breccia, as well as the breccia initially recovered from the mine dump, corresponds to the facies D and E identified in the main pit (*Dirks et al., 2010*). Facies D includes a fossil-rich breccia deposit that contained the fossil material associated with MH2 (*Dirks et al., 2010*; *Val et al., 2018*). Therefore, the geological evidence suggests that the material used for the construction of the miner's road was sourced on-site, and most probably originated from the northern section of the main pit.

The newly discovered vertebrae (second and third lumbar) are preserved in articulation with each other (*Figure 2*, *Figure 2—figure supplement 1*) and refit at multiple contacts with the previously known penultimate (fourth) lumbar vertebra (*Figure 3*). Together, the new and previously known (*Williams et al., 2013*; *Williams et al., 2018b*) vertebral elements form a continuous series from the antepenultimate thoracic vertebra through the fifth sacral element, with only the first lumbar vertebra missing major components of morphology (*Figure 3—figure supplement 1*). The presence of a nearly complete lower back of MH2 allows us to more comprehensively evaluate the functional morphology and evolution of purported adaptations to bipedalism in *A. sediba* and test the hypotheses that the following fundamental features are similar to modern humans (*Homo sapiens*) and distinct from extant great apes: (1) lumbar lordosis, (2) progressive widening of the articular facets and laminae (pyramidal configuration) of the lower back, and (3) overall middle lumbar vertebra shape. Specifically, for these hypotheses, we predict that measurements of combined lumbar wedging (representing degree of lordosis ascertained from available lumbar vertebrae) will fall within the human range (H1), that the configuration of the articular facets and laminae will progressively widen caudally (rather than remaining constant or becoming increasingly narrow) as seen in modern humans (H2), and that the most complete lumbar vertebra of MH2 (U.W.88–233) will fall within the human range of variation in shape analyses (H3).

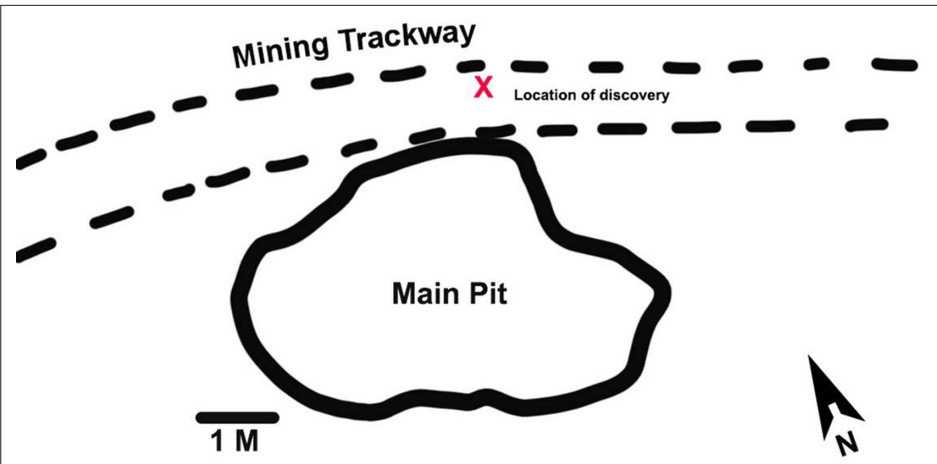

**Figure 1.** Malapa site map showing the location of the new discoveries. The new fossils were discovered during excavations of an early 20th century mining road north of the main pit at Malapa. The location of the block containing the new fossils in the mining trackway is shown with a red X.

## Results

The five new fossils, U.W.88–232, U.W.88–233, U.W.88–234, U.W.88–280, and U.W.88–281, are described below and shown in *Figure 4*. Measurements are included in *Table 1*. A depiction of the anatomical features mentioned in the descriptions below and throughout the manuscript is shown in *Figure 4—figure supplement 1*.

### Descriptions of new fossil material

We determine the seriation of the vertebrae described here based on their direct articulation with one another and refits with previously known vertebrae. Most of the sacrum (U.W.88–137) is preserved in articulation with the neural arch of the last lumbar vertebra (U.W.88–138), which articulates in turn with the inferior portion of the neural arch of the penultimate lumbar vertebra (U.W.88–154). Corresponding vertebral bodies (U.W.88–126 and U.W.88–127, respectively) are preserved together and can be refitted with the neural arches (*Williams et al., 2013*). The new lumbar vertebrae are preserved in partial articulation, including an upper neural arch that refits in two places with U.W.88–154. Therefore, portions of five vertebrae are preserved, followed by a sacrum and preceded by at least three lower thoracic vertebrae (*Williams et al., 2018b*).

U.W.88–280: This is a partial, superior portion of a vertebral body concealed in the matrix of a previously known block containing lower thoracic vertebrae (U.W.88–114, U.W.88–43, and U.W.88–44, antepenultimate, penultimate, and ultimate thoracic vertebrae, respectively, of MH2) (*Williams et al., 2018b*). U.W.88–280 was revealed in the segmentation of micro-CT (hereafter, µCT) data. U.W.88–280 represents the right side of an upper vertebral body with preservation approaching the sagittal midline. The preserved portions measure 16.5 mm dorsoventrally and 14.0 mm mediolaterally at their maximum lengths. The lateral portion of the vertebral body is only preserved ~5.0 mm inferiorly from the superior surface, but there is no indication of a costal facet on the preserved portion. We identify this as part (along with U.W.88–281) of the first lumbar vertebra of MH2 based on its position below the vertebral body of what is almost certainly the last thoracic vertebra (U.W.88–44) (*Williams et al., 2018b*; *Figure 2—figure supplement 1*).

U.W.88–281: This is the partial neural arch of a post-transitional, upper lumbar vertebra concealed in matrix above the subjacent lumbar vertebra (U.W.88–232). It was revealed through the segmentation of µCT data. It consists of the base and caudal portion of the spinous process and parts of the inferior articular processes. The remainder of the vertebra is sheared off and unaccounted for in the block containing the new lumbar vertebrae. U.W.88–281 is fixed in partial articulation with the subjacent second lumbar vertebra (L2), U.W.88–232. Therefore, we identify U.W.88–281 as part of the first lumbar vertebra based on its morphology and position within the block. The left inferior articular facet (IAF) is more complete than the right, with approximately 6.0 mm of its superior-inferior (SI) height

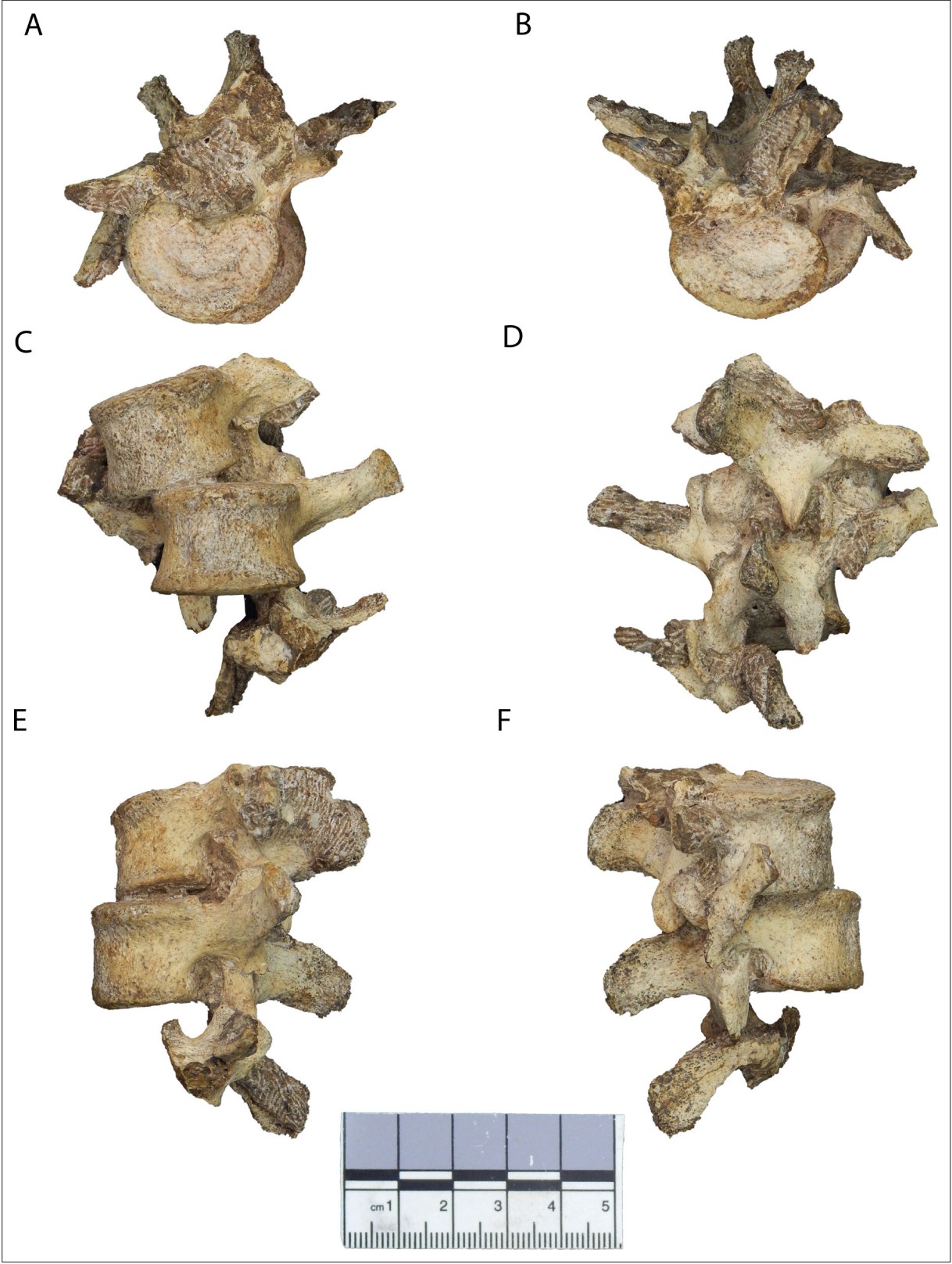

**Figure 2.** New lumbar vertebrae of Malapa Hominin 2 (MH2). Vertebrae in (**A**) superior, (**B**) inferior, (**C**) ventral, (**D**) dorsal, (**E**) left lateral, and (**F**) right lateral views. The partial inferior articular facets of the first lumbar vertebra are embedded in matrix (see *Figure 2—figure supplement 1*). The second lumbar vertebra (U.W.88–232) is in the superior-most (top) position, the third lumbar vertebra (U.W.88–233) is in the middle, and portions of the upper neural arch of the fourth lumbar vertebra (U.W.88–234) are in the inferior-most (bottom) position. These fossils are curated and available for study at the

*Figure 2 continued on next page*

*Figure 2 continued*

University of the Witwatersrand.

The online version of this article includes the following figure supplement(s) for figure 2:

**Figure supplement 1.** Surface renderings of lower thoracic and lumbar vertebrae generated from micro-CT scans.

preserved, and is complete mediolaterally, measuring ~8.0 mm in width. The minimum distance between the IAF is 12.5 mm, and the maximum preserved distance between them is 21.75 mm. The preserved portion of the spinous process is 12.75 mm in dorsoventral length.

U.W.88–232: This vertebra is the L2 and remains in articulation with the third lumbar vertebra (L3), U.W.88–233, held together with matrix. Some portions of U.W.88–232 are covered by adhering matrix or other fossil elements (U.W.88–281 and U.W.88–282, the latter being the sternal end of a clavicle), so μCT data were used to visualize the whole vertebra (*Figure 4*). U.W.88–232 is mostly complete, missing the cranial portions of its superior articular processes and distal portions of its costal (transverse) processes. It is distorted due to crushing dorsally from the right side and related breakage and slight displacements of the left superior articular process at the *pars interarticularis* and the right costal process at its base. Although broken at its base and displaced slightly ventrally, the right costal process is more complete than the left side, which is broken and missing ~10.0 mm from its base. Because of crushing, the neural arch is displaced toward the left side, and the vertebral foramen is significantly distorted. A partial mammillary process is present on the left superior articular process, sheared off along with the remainder of the right superior articular process ~8.0 mm from its base. The left side is similar but much of the mammillary process is sheared off in the same plane as the right side, leaving only its base on the lateral aspect of the right superior articular process. The vertebral body is complete and undistorted, and the spinous process and inferior articular processes are likewise complete but affected by distortion. Standard measurements of undistorted morphologies are reported in *Table 1*.

U.W.88–233: This is the L3 and the most complete vertebra in the lumbar series, although some aspects of the neural arch are distorted, broken, and displaced. It is held in matrix and partial articulation with U.W.88–234, the subjacent partial fourth lumbar vertebra (L4). Due to its position between articulated elements U.W.88–232 and –234 and some adhering matrix, U.W.88–233 was visualized using μCT data. U.W.88–233 is essentially complete; however, like U.W.88–232, the neural arch is crushed from the dorsal direction, with breaks and displacement across the right *pars interarticularis* and the right costal process at its base, with additional buckling around the latter near the base of the of the right superior articular process, resulting in a crushing of the vertebral foramen. The vertebral body, pedicles, spinous process, and superior and inferior articular processes are complete, as are the lamina and costal processes aside from the aforementioned breakage. The left costal process is unaffected by taphonomic distortion. Standard measurements of undistorted morphologies are reported in *Table 1*.

U.W.88–234: This is a partial neural arch of the previously known penultimate lumbar vertebra (L4) (U.W.88-127/153). U.W.88–234 refits in two places with the previously known L4: its partial pedicle with the vertebral body (U.W.88–127) and its spinous process with the inferior base of the spinous process and inferior articular processes (U.W.88–153) (*Figures 2–3*). Only the spinous process and right pedicle, costal process, superior articular processes, and partial lamina are present and in articulation with U.W.88–233. Matrix adheres to the spinous process and costal process, so for this element μCT data were used to visualize and virtually refit it with U.W.88-127/153, forming a partial L4 missing the left superior articular process, costal process, most of the pedicle, the right lateral aspect of the inferior articular process, a portion of the lamina, the inferior aspect of the costal process, and a wedge-shaped area of the lateral body-pedicle border. Preserved standard measurements are reported in *Table 1*.

## Wedging angles and inferred lumbar lordosis

Wedging of articulated vertebrae contribute to the multiple sagittal curvatures of the human spine, with dorsal wedging of lower lumbar vertebrae contributing to a ventrally convex curvature of the lumbar spine (lumbar lordosis). This sinusoidal configuration passively balances the upper body over the pelvis and allows for the unique system of weight bearing and force transmission found in members

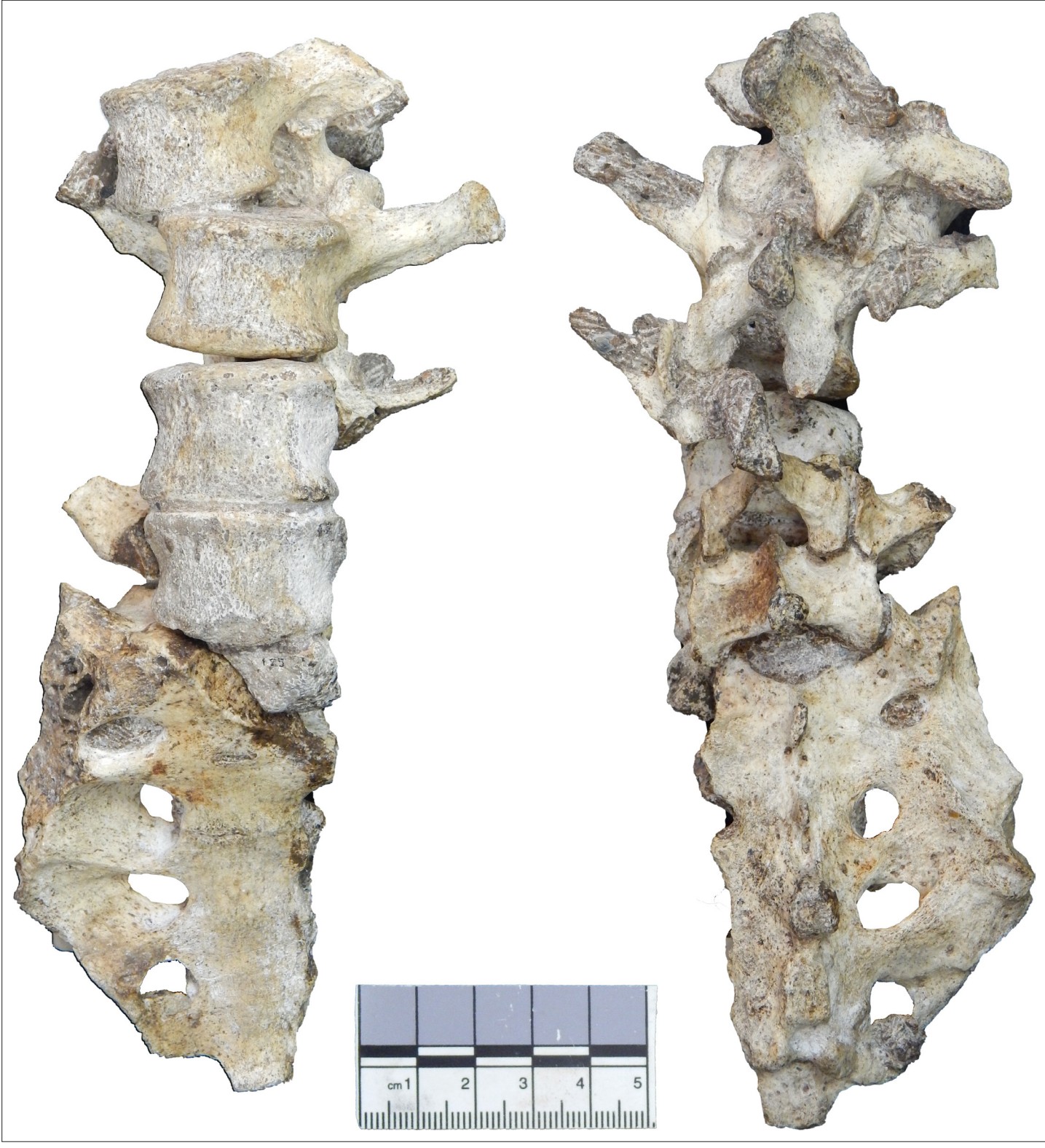

**Figure 3.** The lower back of Malapa Hominin 2 in ventral (left) and dorsal (right) views. New second and third lumbar vertebrae (U.W.88–232, U.W.88–233) are positioned at the top, and U.W.88–234 contributes to the upper portion of the fourth lumbar vertebra (U.W.88–127/153/234). The fifth lumbar vertebra (U.W.88–126/138) sits atop the sacrum (U.W.88–137/125). The lower back elements are preserved together in four blocks, each containing multiple elements held together in matrix and/or in partial articulation: (1) The vertebral body fragment of L1 (U.W.88–280) is preserved within the matrix of a block containing the lower thoracic vertebrae (U.W.88–43/114 and U.W.88–44) (*Figure 2—figure supplement 1*, *Figure 3—figure supplement*

*Figure 3 continued on next page*

*Figure 3 continued*

1); (2) L1 inferior neural arch (U.W.88–281; concealed in matrix), L2 (U.W.88–232), L3 (U.W.88–233), and upper neural arch of L4 (U.W.88–234); (3) the L4 (U.W.88–127) and L5 (U.W.88–126) vertebral bodies, and partial S1 body (U.W.88–125); (4) most of the sacrum (U.W.88–137), the neural arch of L5 (U.W.88–153), the inferior portion of the neural arch of L4 (U.W.88–138).

The online version of this article includes the following figure supplement(s) for figure 3:

**Figure supplement 1.** Surface renderings of the new lumbar vertebrae positioned between previously known vertebrae.

of the human lineage (**Davis, 1961**; **Robinson, 1972**; **Pal and Routal, 1987**; **Latimer and Ward, 1993**; **Shapiro, 1993a**; **Lovejoy, 2005**; **Whitcome et al., 2007**; **Masharawi et al., 2010**; **Been et al., 2014**; **Tardieu et al., 2017**). Wedging angles for individual lumbar vertebrae (L2-L5) and combined L2-L5 wedging were calculated for *A. sediba* and the comparative sample and are presented in (**Figure 5**, **Figure 5—figure supplement 1**, **Figure 5—figure supplement 2**) and **Table 2** and **Table 3**. MH2 possesses the greatest (i.e., most negative) combined wedging value of any adult early hominin (–6.8°). Although all fossil hominins fall within the 95% prediction intervals of modern humans, only MH2 falls outside the 95% prediction intervals of great apes in combined L2-L5 wedging (**Figure 5**).

Patterns of change across lumbar levels demonstrate that MH2's vertebrae transition from ventral (kyphotic) to dorsal (lordotic) wedging between the L3 and L4 levels; however, all adult fossil hominins fall within the 95% prediction intervals of modern humans (**Figure 5**, **Figure 5—figure supplement 1**). As shown previously (**Williams et al., 2013**), the last lumbar vertebra of MH2 is strongly dorsally wedged like that of the Kebara 2 Neandertal and the juvenile specimen KNM-WT 15000, whereas other fossil hominins do not demonstrate this pattern. Although vertebral wedging is characterized by high levels of variation within groups, especially in combined L2-L5 wedging (**Figure 5**, **Table 2**), the pattern of lumbar wedging angles observed in MH2 (i.e., transition from penultimate to ultimate lumbar level) and its combined L2-L5 wedging fall within the modern human 95% PIs and outside those of great apes (**Figure 5**, **Figure 5—figure supplement 1**, **Figure 5—figure supplement 2**). The hypothesis that *A. sediba* is human-like in lumbar wedging, therefore, cannot be rejected.

## Configuration of the neural arch

The recovery of new lumbar vertebrae of MH2 allows for the quantification and comparison of inter-articular facet width increase in *A. sediba*. Humans are characterized by a pyramidal configuration of the articular facets such that they increase in transverse width progressively down the lumbar column (i.e., from cranial to caudal) (**Latimer and Ward, 1993**; **Ward and Latimer, 2005**). Using an index of the last lumbar-sacrum inter-articular maximum distance relative to that of lumbar vertebrae three levels higher (L2-L3 in hominins, L1-L2 in chimpanzees and gorillas), we show that *Australopithecus africanus* (Sts 14 and StW 431; average = 1.42) and *A. sediba* (1.43) fall at the low end of the range of modern human variation in this trait (**Figure 6**). We note that A.L. 288–1 (*Australopithecus afarensis*) falls at the low end of human variation near other australopiths if the preserved lumbar vertebra (A.L. 288-1aa/ak/al) is treated as an L3 (**Latimer and Ward, 1993**; **Lovejoy, 2005**; **Johanson et al., 1982**; **Meyer et al., 2015**), but outside the range of human variation and within that of orangutans if it is treated as an L2 (**Cook et al., 1983**). *Homo erectus* and Neandertals fall well within the range of modern human variation. The presence of a pyramidal configuration of the lumbar articular facets is therefore present in MH2, supporting our hypothesis that *A. sediba* was adapted to a human-like configuration of the neural arch.

## Middle lumbar vertebra (L3) comparative morphology

The new middle lumbar vertebra, U.W.88–233, is complete, and although the neural arch is compressed ventrally into the vertebral foramen space, it can be reasonably reconstructed from μCT data (see Materials and methods). We used three-dimensional geometric morphometrics (3D GM) to evaluate the shape affinities of U.W.88–233 among humans, great apes, and fossil hominins. The results of our principal components analysis (PCA) on Procrustes-aligned shape coordinates reveal that *A. sediba* falls within or near the human distribution on the first three principal components (PC1–3) (**Figure 7**). PC1 explains 31% of the variance in the dataset, and along it hominins are characterized by more sagittally oriented and concave superior articular facets (SAF), more dorsally oriented costal processes, a dorsoventrally shorter and cranially oriented spinous process, craniocaudally shorter,

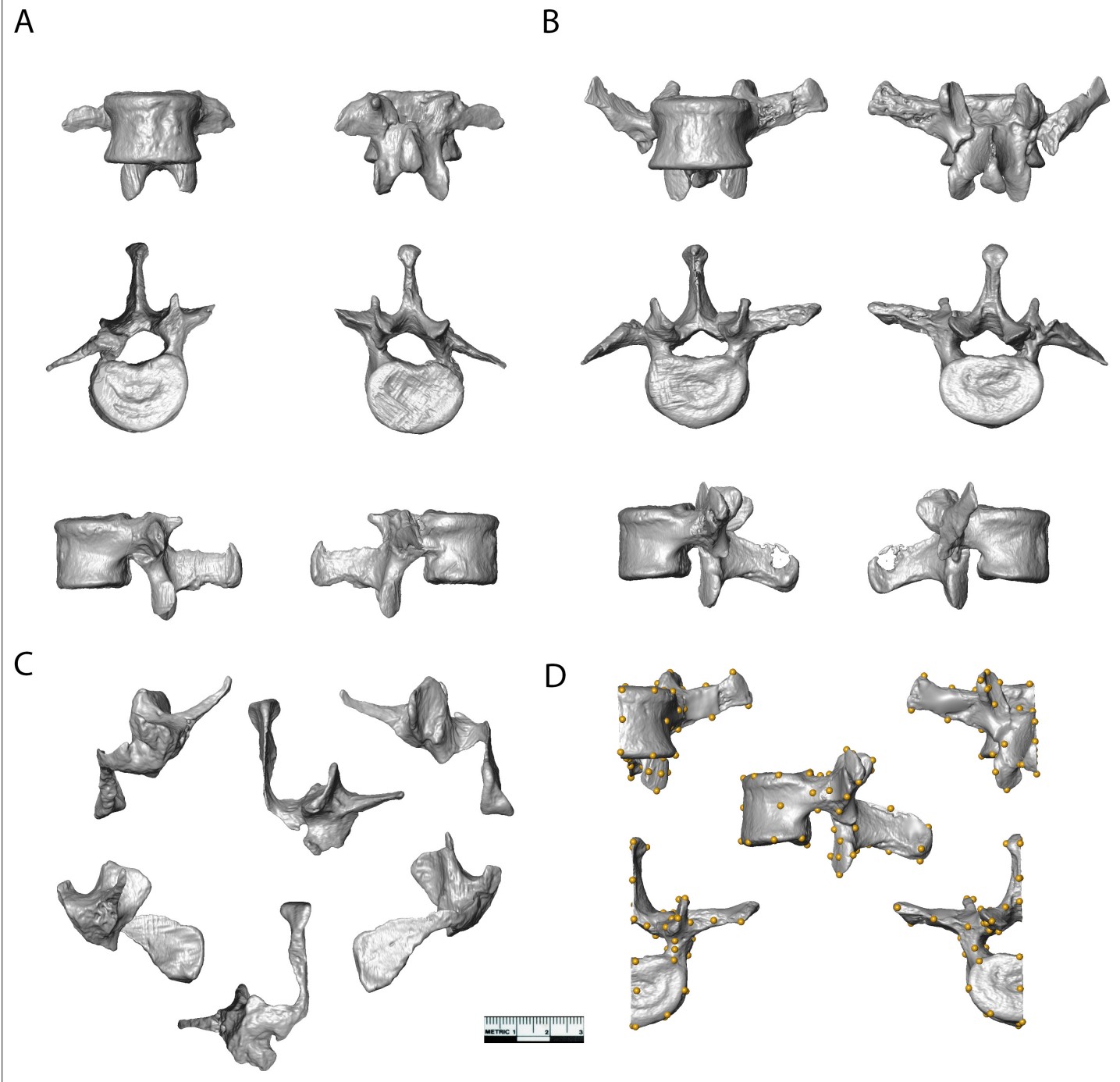

**Figure 4.** Surface models of vertebrae from the new lumbar block. (**A**) U.W.88–232 (L2) and (**B**) U.W.88–233 (L3) shown in ventral (top left), dorsal (top right), superior (middle left), inferior (middle right), left lateral (bottom left), and right lateral (bottom right) views. (**C**) U.W.88–234 (L4) in ventral (top left), dorsal (top right), superior (top middle), left lateral (bottom left), right lateral (bottom right), and inferior (bottom middle) views. (**D**) Left half of U.W.88–233 showing the 48 landmarks used in the three-dimensional geometric morphometric (3D GM) analyses.

The online version of this article includes the following figure supplement(s) for figure 4:

**Figure supplement 1.** Anatomical features of lumbar vertebrae.

dorsoventrally longer vertebral body, and more caudally positioned SAF and IAF relative to the vertebral body compared to great apes.

PC2 explains 13% of the variance and contrasts long spinous processes and relatively neutrally wedged (~0° ± 1°) vertebral bodies of hominins and African apes with the shorter spinous processes

**Table 1.** Measurements on lumbar vertebrae in mm for linear data and degrees for angles (measurement definitions are included in the supplementary material).

| | U.W.88–232 (L2) | U.W.88–233 (L3) | U.W.88-127/ 153/234 (L4) | U.W.88-126/138 (L5) |
|---|---|---|---|---|
| 1. Body sup. transv. width | 29.5 | 30.1 | 31.4 | 32.8 |
| 2. Body sup. dorsoven. dia. | 20.8 | 21.4 | 22.2 | 21.4 |
| 3. Body inf. transv. width | 29.0 | 31.4 | 32.4 | 28.8 |
| 4. Body inf. dorsoven. dia. | 21.1 | 21.0 | 21.2 | 19.8 |
| 5. Body ventral height | 21.0 | 21.75 | 22.1 | 21.0 |
| 6. Body dorsal height | 22.5 | 22.25 | 21.5 | 17.0 |
| 7. Body wedging angle (calculated) | 4.1° | 1.3° | –1.6° | –10.7° |
| 8. Vertebral foramen dorsoven. dia. | 10.5 | 8.85 | – | 23.0 |
| 9. Vertebral foramen transv. dia. | 17.6 | 17.3 | – | 16.3 |
| 10. Sup.-inf. inter-AF height | – | 37.0 | 32.6 | 31.5 |
| 11. Max. inter-SAF dist. | – | 24.0 | – | 28.5 |
| 12 . Min. inter-SAF dist. | – | 14.5 | – | – |
| 13. Max. inter-IAF dist. | 23.0 | 25.0 | (28.0)* | (33.0) |
| 14 . Min. inter-IAF dist. | 11.0 | 9.5 | 11.6 | 15.6 |
| 15. SAF sup.-inf. dia. | – | 12.8 | – | 13.4 |
| 16. SAF transv. dia. | – | 11.5 | – | 10.8 |
| 17. IAF sup.-inf. dia. | 11.5 | 11.5 | 14.7 | 14.4 |
| 18. IAF transv. dia. | 8.1 | 8.9 | 9.2 | 11.7 |
| 19. Spinous process angle | 176° | 160° | 163° | 166° |
| 20. Spinous process length | 27.0 | 28.0 | 28.0 | 23.6 |
| 21. Spinous process terminal trans. width | 6.9 | 7.4 | 8.1 | 6.85 |
| 22. Spinous process terminal sup.-inf. height | 13.8 | 11.75 | 12.7 | 7.15 |
| 23. Costal process base sup.-inf. height | 11.5 | 12.2 | – | 13.9 |
| 24. Costal process angle | 78° | 82° | – | 50° |
| 25. Costal process length | – | 31.0 | – | – |
| 26. SAF orientation (in degrees) | – | 31° | 33° | 26° |
| 27. Pedicle sup.-inf. height | 10.9 | 10.6 | – | 11.2 |
| 28. Pedicle transv. width | 5.9 | 7.1 | 9.0 | 10.9 |
| 29. Pedicle dorsoven. length | 5.0 | 5.6 | 6.5 | 7.0 |
| 30. Lamina sup.-inf. height | 16.1 | 15.4 | – | 14.0 |
| 31. Lamina transv. width | 20.0 | 22.0 | – | 30.5 |

*Parentheses indicate that the structure is incomplete and its measurement if estimated.

and strongly ventrally wedged vertebral bodies of orangutans. PC3 explains 8% of the variance and largely contrasts dorsoventrally longer vertebral bodies with caudally oriented spinous processes in gorillas with dorsoventrally shorter vertebral bodies and less caudally oriented spinous processes in chimpanzees and orangutans; hominins fall intermediate between these groups.

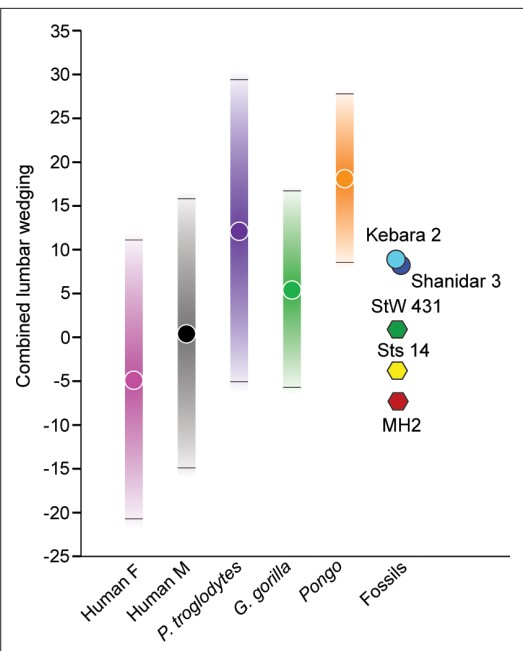

**Figure 5.** Combined L2-L5 vertebral body wedging angles. Lumbar vertebral body wedging angles are summed from levels L2 through L5. Only fossil specimens preserving the last four lumbar vertebrae are included (*Australopithecus africanus*: Sts 14, StW 431; *Australopithecus sediba*: MH2; Neandertals: Kebara 2, Shanidar 3). For the extant taxa, 95% prediction intervals are shown with bars. Table 2 includes summary statistics, Table 3 fossil hominin data, and *Figure 5— source data 1* provides the raw data.

The online version of this article includes the following source data and figure supplement(s) for figure 5:

**Source data 1.** Lumbar wedging angles and combined wedging angles values of extant taxa (Excel file).

**Figure supplement 1.** Lumbar vertebral body wedging angles in fossil hominins and modern humans.

**Figure supplement 2.** Lumbar vertebral body wedging angles in fossil hominins and extant great apes.

PC4 explains 5% of the variance, and contrasts *A. sediba* and *A. africanus* with both humans and great apes. Sts 14 and especially U.W.88–233 are characterized by longer, taller, more cranially oriented costal processes that do not taper distally and more sagittally oriented (as opposed to more coronally oriented) articular facets (*Figure 7*, *Figure 7—figure supplement 1*). We removed great apes and reran the PCA to ensure that their presence is not affecting the relationship of fossil hominins to modern humans. This hominin-only PCA essentially reproduced the results of PC4 on PC2 (*Figure 7—figure supplement 2*), confirming that Sts 14 and U.W.88–233 are distinct from modern humans in costal process morphology. Therefore, although the *A. sediba* middle lumbar vertebra is somewhat human-like in overall shape, its vertebral body is intermediate in shape between great apes and modern humans and its costal processes are robust, cranially oriented, and cranially positioned on the pedicles.

To provide a more in-depth comparison of the morphometric affinities of U.W.88–233, we plot Procrustes distances between U.W.88–233 and Sts 14, Shanidar 3, the modern human sample, and the chimpanzee sample (*Figure 7—figure supplement 1A*). We also show pairwise comparisons of Procrustes distances for middle lumbar vertebra shape within both human and chimpanzee samples, and between human and chimpanzee samples (*Figure 7—figure supplement 1C*). These analyses demonstrate that U.W.88–233 is most similar to Sts 14 among fossil and extant specimens included.

To include other fossil hominins with broken processes, we ran a second 3D GM analysis excluding the majority of costal process and spinous process landmarks. This analysis, which includes landmarks on the vertebral body, SAF and IAF, and the bases of the costal and spinous processes, produces a similar pattern compared to the analysis on the full landmark set (*Figure 7*). Humans and great apes separate along PC1, which is largely explained by vertebral body heights (including vertebral wedging) and SI position of the articular facets relative to the vertebral body. U.W.88–233, like other early hominins included in this study, falls intermediately between modern humans and great apes along PC1.

We used a Procrustes distance-based analysis of variance (ANOVA) to evaluate the effect of centroid size on lumbar shape (*Goodall, 1991*). The results show significant effects of centroid size ($F$ = 9.83; $p < 0.001$), genus (with hominins pooled; $F$ = 27.7; $p < 0.001$), and an interaction between genus and centroid size ($F$ = 1.48; $p = 0.01$), implying unique shape allometries within genera (*Table 4*). We plotted standardized shape scores derived from a multivariate regression of shape on centroid size against centroid size to visualize shape changes (*Drake and Klingenberg, 2008*; *Figure 7—figure supplement 3*). In general, larger centroid sizes are associated with 3D shape changes including dorsoventrally longer and more caudally projecting spinous processes, more cranially oriented and less sagittally oriented costal processes, and less caudally projecting IAF. Importantly, however, the cranially oriented costal processes of U.W.88–233 (and Sts 14) appear not to be explained by centroid

**Table 2.** Summary statistics for lumbar wedging angles of the extant comparative sample.

| Level | Group/fossil | Human ♂ (48) | Human ♀ (31) | *Pan* (43) | *Gorilla* (31) | *Pongo* (10) |
|---|---|---|---|---|---|---|
| L2 | Mean (stdev) | 4.4 (3.3) | 2.1 (2.1) | 4.6 (3.0) | 2.3 (2.7) | 5.3 (4.7) |
| | 95% PI | –2.1, 10.9 | –2.0, 6.2 | –1.3,10.5 | –3.0, 7.6 | –3.9, 14.5 |
| | Min, max | –3.4, 12.8 | –2.1, 6.0 | –1.1, 12.4 | –4.0, 8.3 | –0.4, 14.4 |
| L3 | Mean (stdev) | 2.4 (3.1) | 1.3 (2.6) | 4.5 (3.1) | 2.6 (2.0) | 6.1 (2.0) |
| | 95% PI | –3.7, 8.5 | –3.8, 6.4 | –1.6, 10.6 | –1.3, 6.5 | 2.2, 10.0 |
| | Min, max | –4.8, 11.2 | –3.4, 6.6 | –2.4, 10.9 | –2.2, 7.5 | 3.2, 8.6 |
| L4 | Mean (stdev) | –0.5 (2.8) | –1.5 (2.9) | 3.5 (3.4) | 1.3 (2.5) | 4.8 (3.8) |
| | 95% PI | –6.0, 5.0 | –7.2, 4.2 | –3.2, 10.2 | –3.6, 6.2 | –2.6, 12.2 |
| | Min, max | –8.3, 4.8 | –7.9, 4.6 | –3.6, 10.9 | –3.9, 7.1 | –1.0, 11.3 |
| L5 | Mean (stdev) | –5.9 (2.9) | –6.5 (3.1) | –0.5 (2.8) | –0.8 (2.2) | 2.0 (4.1) |
| | 95% PI | –11.6,–0.2 | –12.6,–0.4 | –6.0, 5.0 | –5.1, 3.5 | –6.0, 10.0 |
| | Min,max | –11.6, 1.8 | –12.3, 2.2 | –5.9, 7.0 | –6.2, 3.5 | –5.2, 8.0 |
| Sum | Mean (stdev) | 0.4 (7.8) | –4.9 (8.1) | 12.1 (8.8) | 5.4 (5.7) | 18.1 (9.5) |
| | 95% PI | –15.0, 15.7 | –20.8, 11.0 | –5.1, 29.3 | –5.8, 16.6 | 8.6, 27.6 |
| | Min, max | –15.6, 15.5 | –18.2, 11.2 | –5.2, 31.7 | –8.1, 15.7 | 0.04, 36.0 |

size given its relatively small size and overlap with *Pan* in standardized shape scores (*Figure 7—figure supplement 3*).

## Discussion

The recovery of two new lumbar vertebrae and portions of other lumbar vertebrae of the adult female *A. sediba* (MH2), together with previously known vertebrae, form a nearly complete lumbar column (*Figure 3*, *Figure 3—figure supplement 1*) and allows us to test hypotheses based on more limited

**Table 3.** Lumbar wedging angles and combined wedging of fossil hominin specimens.

| | L2 | L3 | L4 | L5 | Combined |
|---|---|---|---|---|---|
| Kebara 2 | 8.1 | 6.9 | 4.5 | –10.6 | 8.9 |
| Shanidar 3 | 8.0 | 5.1 | 0.1 | –4.9 | 8.3 |
| La Chapelle-aux-Saints 1 | – | 4.7 | 0.0 | –7.8 | – |
| KNM-WT 15000 | – | – | –8.3 | –11.8 | – |
| LES1 | 3.0 | – | – | – | – |
| SK 3981b | – | – | – | –3.5 | – |
| MH2 | 4.1 | 1.4 | –1.6 | –11.2 | –7.3 |
| Sts 14 | 2.3 | 1.7 | –0.9 | –6.9 | –3.8 |
| StW 431 | 2.0 | 2.3 | 0.9 | –4.2 | 1.1 |
| StW 8 | 5.2 | 3.6 | – | – | – |
| StW 572 | 4.8 | – | – | – | – |
| StW 656/600 | – | 4.2 | – | –6.2 | – |
| A.L. 288–1 | – | 7.2 | – | – | – |
| A.L. 333–73 | – | 2.8 | – | – | – |

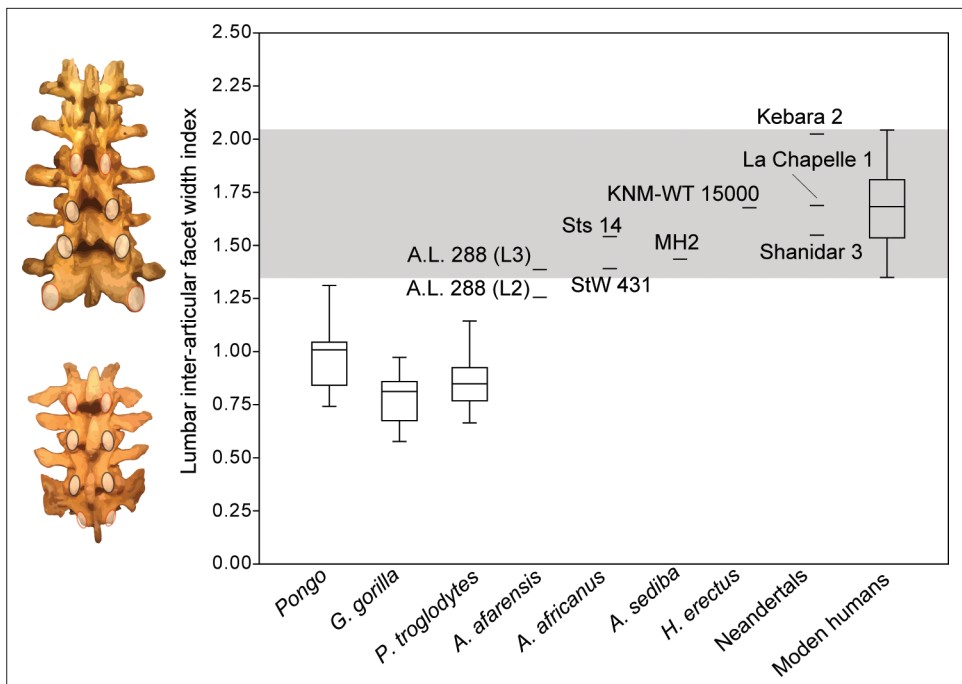

**Figure 6.** Pyramidal configuration of articular facet spacing in hominids. The inter-articular facets of the last lumbar/sacrum and those of lumbar vertebrae three elements higher in the column (L1-L2 in chimpanzees and gorillas with four lumbar vertebrae; L2-L3 in hominins) are included as the numerator and denominator, respectively, in a lumbar inter-articular facet index. These levels are highlighted on the left in red in both a human (top) and a chimpanzee (bottom). The gray box highlights the range of variation observed in the modern human sample. All great apes are significantly different from modern humans (<0.001). The ratio data for inter-articular facet spacing can be found in *Figure 6—source data 1*.

The online version of this article includes the following source data for figure 6:

**Source data 1.** Inter-articular facet ratios of fossil hominins and extant taxa (Excel file).

material. As we outline below, *A. sediba* demonstrates evidence for lumbar lordosis in the combined pattern of bony wedging of lumbar vertebral bodies, as well as progressive widening of neural arch structures moving caudally ('pyramidal configuration') of lumbar vertebrae and the sacrum, which does not allow us to reject the hypotheses that *A. sediba* has human-like adaptations to bipedalism. However, the hypothesis that *A. sediba*'s middle lumbar vertebra (L3) is human-like is not fully supported: although U.W.88–233 is somewhat human-like in overall shape, its costal processes are long and cranially oriented, unlike modern humans, and its vertebral body is intermediate in shape between those of modern humans (and Neandertals) and great apes.

*Williams et al., 2013*, predicted strong lumbar lordosis ('hyperlordosis') in MH2 based on the combined wedging values of the penultimate and ultimate lumbar vertebrae. In contrast, *Been et al., 2014*, estimated lumbar lordosis angle using pelvic incidence from a pelvis reconstruction (*Kibii et al., 2011*) and found MH2 to produce the least lordotic lumbar column of the sampled members of the genus *Australopithecus* in their sample, falling well below modern human values and within the distribution of Neandertals. Neandertals are thought to be 'hypolordotic', or characterized by a relatively straight, non-lordotic lumbar column (*Been et al., 2014*; *Been et al., 2017*; but see *Haeusler et al., 2019*). However, *Tardieu et al., 2017*, report a human-like degree of pelvic incidence (and therefore lumbar lordosis) in a new reconstruction of the MH2 pelvis. Therefore, current interpretations of lumbar curvature of *A. sediba* range from hyperlordotic to hypolordotic. Here, we report that the pattern of vertebral wedging of MH2 and most other fossil hominins are similar to both modern humans and extant great apes except at the last lumbar level, where MH2 is markedly more dorsally wedged (*Figure 5*, *Figure 5—figure supplement 1*, *Figure 5—figure supplement 2*). Like the Neandertal Kebara 2, the strong dorsal (lordotic) wedging of MH2's last lumbar vertebra is likely countering a strong ventral (kyphotic) wedging in the upper lumbar column (*Figure 5*). However, MH2

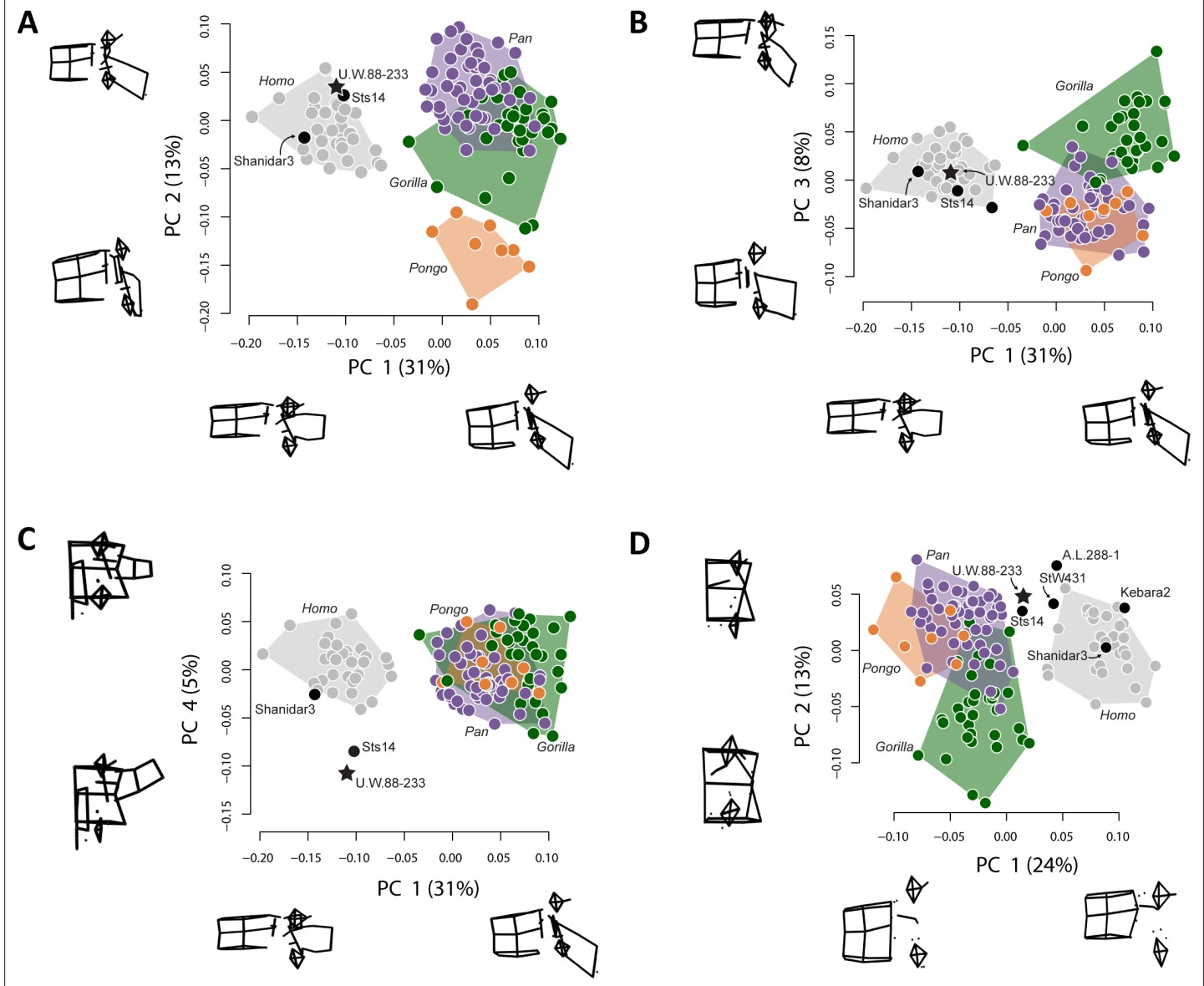

**Figure 7.** Principal components analysis (PCA) on middle lumbar vertebra three-dimensional (3D) landmark data. (**A–C**) PCA on the full set of 48 landmarks, including Sts 14 (*Australopithecus africanus*), U.W.88–233 (*Australopithecus sediba*), and Shanidar 3 (Neandertal). (**A–B**) Hominins separate from great apes on PC1 (wireframes in lateral view), African apes and hominins separate from orangutans on PC2 (wireframes in lateral view), and (**C**) *Australopithecus* species separate from other hominids on PC4 (wireframes in posterior view). Note that spinous and costal process lengths and orientations drive much of the variance in middle lumbar vertebrae. (**D**) PCA on a reduced landmark set (excluding spinous and costal process landmarks) to include A.L. 288–1 (*Australopithecus afarensis*), StW 431 (*A. africanus*), and Kebara 2 (Neandertal). Notice that *Australopithecus* specimens fall outside the modern human convex hulls, with Sts 14 and MH2 close to those of the African apes. 3D landmark data were subjected to Procrustes transformation.

The online version of this article includes the following figure supplement(s) for figure 7:

**Figure supplement 1.** Procrustes distances and mean differences from U.W.88–233.

**Figure supplement 2.** Middle lumbar morphology of *Australopithecus sediba* and other hominins.

**Figure supplement 3.** The effect of middle lumbar centroid size on shape.

demonstrates much less ventral wedging than Neandertals and produces a human-like combined wedging angles value, falling outside the 95% prediction intervals of great apes. Therefore, it seems likely that MH2 and possibly the juvenile *H. erectus* individual KNM-WT 15000 demonstrate strong dorsal wedging at the last lumbar level for different reasons than Kebara 2. It was suggested previously

**Table 4.** Procrustes analysis of variance (ANOVA) results of centroid size and middle lumbar vertebra shape.

| | Df | SS | MS | $R^2$ | F | Z | Pr (>F) |
|---|---|---|---|---|---|---|---|
| Centroid size | 1 | 0.11917 | 0.11917 | 0.0452 | 9.8252 | 5.5237 | <0.0001 |
| Genus | 3 | 1.00793 | 0.33598 | 0.38234 | 27.7009 | 12.0794 | <0.0001 |
| Centroid size:genus | 3 | 0.0537 | 0.0179 | 0.02037 | 1.4759 | 2.2858 | 0.0109 |
| Residuals | 120 | 1.45545 | 0.01213 | 0.55209 | | | |
| Total | 127 | 2.63626 | | | | | |

that the morphology of the MH2 last lumbar is part of a kinematic chain linked to hyperpronation of the foot (*DeSilva et al., 2013*; *Williams et al., 2013*). With the absence of soft tissue contributions to the kinematic chain (i.e., intervertebral discs), formal biomechanical testing is beyond the scope of the current paper; however, our results suggest that MH2 was probably neither hypolordotic nor hyperlordotic and produces a combined wedging angles value more similar to modern humans than great apes.

Modern humans are characterized by a pyramidal configuration of the lumbar inter-articular facet joints such that the upper lumbar articular facet joints (and associated laminae) are transversely more narrowly spaced than those of the lower lumbar vertebrae and especially compared to the lumbosacral inter-articular facet joints (*Latimer and Ward, 1993*; *Sanders, 1998*; *Ward and Latimer, 2005*). Together with vertebral body and intervertebral disc wedging, this progressive widening facilitates the adoption of lordotic posture during ontogeny and allows for the imbrication of the IAF of a superjacent vertebra onto the laminae of the subjacent vertebra during hyperextension of the lower back (*Latimer and Ward, 1993*; *Williams et al., 2013*). Like modern humans, known fossil hominin lumbar vertebrae bear 'imbrication pockets', mechanically induced fossae positioned just caudal to the SAF on the lamina (*Latimer and Ward, 1993*; *Williams et al., 2013*), providing direct evidence for lumbar hyperextension and lordosis. Inadequate spacing of lower lumbar inter-articular facets in modern humans can result in spondylolysis, fracture of the *pars interarticularis*, and potential separation of the affected vertebra's spinous process and inferior articular processes (*Ward and Latimer, 2005*). Lack of the progressive widening of inter-articular facets of lower lumbar vertebrae in our closest living relatives, the African apes, begs the question of when the pyramidal configuration evolved and to what extent various fossil hominins demonstrated this trait. Although a human-like pattern of interfacet distance was once claimed for the European late Miocene ape *Oreopithecus* (*Köhler and Moya-Sola, 1997*), *Russo and Shapiro, 2013*, demonstrated that measures for changes in both interfacet distance and laminar width in this extinct ape fall within ranges of extant apes (and other suspensory mammals) and outside those of humans. Among hominins, *Latimer and Ward, 1993*, documented the presence of a pyramidal configuration in *H. erectus*. It has also been demonstrated qualitatively in *A. afarensis* and *A. africanus* (*Robinson, 1972*; *Lovejoy, 2005*), and its presence in *A. sediba* could be inferred previously based on the articulated penultimate and ultimate lumbar vertebrae and sacrum of MH2 (*Williams et al., 2013*). Here, we show that MH2 and other *Australopithecus* specimens fall at the low end of modern human variation and differ from great apes in having significantly wider inter-articular facets at the lumbosacral junction than higher in the lumbar column (*Figure 6*).

The overall morphologies of lumbar vertebrae are informative with regard to locomotion and posture in primates (*Slijper, 1946*; *Shapiro, 1993b*; *Sanders and Bodenbender, 1994*; *Granatosky et al., 2014*; *Williams and Russo, 2015*). Hominoids are characterized by derived vertebral morphologies related to orthogrady and antipronograde positional behaviors, and early hominins have been found to largely resemble modern humans in lumbar vertebra shape, with some retained primitive morphologies (*Robinson, 1972*; *Schmid, 1991*; *Shapiro, 1993a*). Our 3D GM results show that the middle lumbar vertebra of *A. sediba* (U.W.88–233; L3) falls with modern humans (L3) to the exclusion of great apes (L2) in overall shape (*Figure 7A–B*). However, it bears long, cranially and ventrally oriented costal processes unlike those of modern humans (*Figure 7C*, *Figure 7—figure supplement 1*, *Figure 7—figure supplement 2*), and the vertebral body is somewhat intermediate in shape between modern humans and great apes (*Figure 7D*).

To evaluate the potential effect of centroid size in driving differences in middle lumbar vertebra shape, we ran a Procrustes distance-based ANOVA on generalized Procrustes analysis (GPA) shape scores to test whether shape differences between this fossil specimen and any extant taxon, such as *H. sapiens*, could be explained by differences in size ('allometry'). Since body mass scales as the cube of linear dimensions and the physiological cross-sectional area of skeletal muscle – a major determinant of isometric force production – scales as the square of linear dimensions, larger-bodied individuals should be relatively weaker with all else held equal. Thus to compensate, we would expect to see changes in bony morphology based on differences in body size. We find that the spinous and costal processes are longer in specimens with larger centroid sizes (*Figure 7—figure supplement 3*). These changes would increase the moment arms of the *erector spinae* and *quadratus lumborum* muscles, respectively, resulting in greater moments that contribute to lumbar extension, ventral flexion, and lateral flexion to cope with increases in body mass. Our results suggest that, while we detect a statistically significant effect of centroid size on middle lumbar vertebral shape within each group, the differences in costal process size and orientation observed between *A. sediba* and modern humans appear not to be explained by size alone.

Long costal processes give the *psoas major* and *quadratus lumborum* muscles an effective leverage in acting on the vertebral column, increasing their moment arms and torque generation capabilities to assist the *erector spinae* in lateral flexion of the spine, back extension, and stabilizing the trunk during upright posture and bipedalism and ape-like vertical climbing (*Robinson, 1972*; *Waters and Morris, 1972*; *Schmid, 1991*; *Sanders, 1998*; *Figure 7—figure supplement 2*). *Psoas major* acts with *iliacus* as a powerful flexor of the thigh and trunk, while *quadratus lumborum* is a trunk extensor and a lateral flexor of the vertebral column and pelvis unilaterally (*Robinson, 1972*; *Drake et al., 2019*). The lumbar vertebral morphology of *A. sediba*, therefore, is that of a biped equipped with especially powerful trunk musculature for stabilizing the hip and back during walking and/or vertical climbing. Further work on back morphology and function in *A. sediba* and other early hominins is required to explore the efficacy of these possible functional explanations for the observed morphology of MH2's lumbar vertebrae.

Previous work has shown that the adult, presumed female individual from Malapa (MH2) demonstrates clear adaptations to bipedal locomotion (*Zipfel et al., 2011*; *DeSilva et al., 2013*; *DeSilva et al., 2018*; *Williams et al., 2013*; *Williams et al., 2018b*), as do other *Australopithecus* specimens, despite their retention of features linked to suspensory behavior and other arboreal proclivities (*Zipfel et al., 2011*; *Henry et al., 2012*; *Churchill et al., 2013*; *Churchill et al., 2018*; *DeSilva et al., 2013*; *DeSilva et al., 2018*; *Prang, 2015a*; *Prang, 2015b*; *Prang, 2016*; *Meyer et al., 2017*; *Rein et al., 2017*; *Holliday et al., 2018*; *Prabhat et al., 2021*). The new fossils here reinforce these conclusions, signaling a lower back in MH2 as that of an upright biped equipped with powerful trunk musculature potentially used in both terrestrial and arboreal locomotion. The recovery and study of new fossil material, including juvenile material such that the ontogeny of bipedal features can be examined (*Ward et al., 2017*; *Nalley et al., 2019*), along with experimental biomechanical work and additional comparative analyses, will allow for testing hypotheses of form and function in the hominin fossil record.

## Materials and methods
### Wedging angle and neural arch configuration
Original fossil material was studied in all cases with the exception of two Neandertal specimens (Kebara 2 and Shanidar 3), for which high-quality casts were used. The *A. sediba* fossils belonging to MH2 (U.W.88-280/281, L1; U.W.88–232, L2; U.W.88–233, L3; U.W.88-127/153/234, L4; U.W.88-126/138, L5) were studied at the University of the Witwatersrand (Johannesburg), as was LES1 *Homo naledi* (U.W.102a-154B, L1) and fossils purportedly belonging to *A. africanus*: StW 431 (StW 431 r, L1; StW 431 s, L2; StW 431t, L3; StW 431 u, L4; StW 431 v, L5), StW 8 (StW8a, L1; StW8b, L2; StW8c, L3; StW8d, L4), StW 572 (L2), StW 656 (L3), and StW 600 (L5). The *A. africanus* specimen Sts 14 (Sts 14e, L1; Sts 14d, L2; Sts 14 c, L3; Sts 14b, L4; Sts 14 a, L5) and possible *Paranthropus robustus* or early *Homo* specimen SK 3981b (L5) were studied at the Ditsong National Museum of Natural History, *A. afarensis* specimens (A.L. 288-1aa/ak/al, L3; A.L. 333–73, L3) at the National Museum of Ethiopia, the

*H. erectus* juvenile individual KNM-WT 15000 (AV/AA, L1; Z/BW, L2; AB, L3; BM, L4; AC, L5) at the National Museums of Kenya, and La Chapelle-aux-Saints 1 at Musée de l'Homme (Paris).

Our comparative sample consisted in total of 43 chimpanzees (*Pan troglodytes*), 31 western gorillas (*Gorilla gorilla*), 14 orangutans (*Pongo* sp.), and 54 modern humans (*H. sapiens*). To ensure that adequate space between elements was taken into account, we only included great apes with four lumbar vertebrae. Eastern gorillas (*Gorilla beringei*), which mostly possess just three lumbar vertebrae (*Williams et al., 2019*), are not included here, nor are other great ape individuals with only three lumbar vertebrae. The human sample includes data from an archaeological sample representing individuals from Africa, Asia-Pacific, and South America studied at the American Museum of Natural History (New York City), Musée de l'Homme, the Natural History Museum (London), and the University of the Witwatersrand. Measurements (listed in Appendix 1) were collected with Mitutoyo digital calipers (Mitutoyo Inc, Japan) and recorded at 0.01 mm; however, we report measurements at 0.1 mm.

Following *Digiovanni et al., 1989*, we calculated wedging angles for lumbar vertebrae 2–5 using the arctangent of difference between the dorsal and ventral height of the vertebral body and its dorsoventral length (see Appendix 1). We also summed those values into a combined lumbar wedging value. For both great apes and male and female humans, 95% prediction intervals of the mean (1.96 * standard deviation) were calculated for each vertebral level and for the combined wedging values.

Inter-articular facet spacing was measured across the lateral borders of the IAF of lumbar vertebrae three levels apart: on the last lumbar vertebra and on L1 in great apes with four lumbar vertebrae and on L2 in hominins. This is done to estimate the difference in inter-articular facet width at upper and lower lumbar levels and thus quantify neural arch configuration. Due to preservation, this measurement was estimated from the SAF of the L3 vertebra and/or the sacrum in a selection of fossils (A.L. 288–1, Sts 14). In instances of partial preservation, the relevant adjacent elements were articulated to estimate the measurement (MH2, StW 431; KNM-WT 15000). An index was created by dividing the last lumbar-sacrum interarticular facet mediolateral width by that of the upper lumbar vertebrae.

## 3D reconstruction and geometric morphometric analysis

For 3D GM analyses, we used subsets of middle lumbar vertebrae that were scanned at the aforementioned institutions using an Artec Space Spider 3D scanner (Source Graphics, Anaheim, CA). The middle lumbar vertebra of hominins with five lumbar vertebrae is the third lumbar vertebra, and that of chimpanzees and gorillas with three lumbar vertebrae is L2. Many chimpanzees and bonobos, western gorillas, and orangutans have four lumbar vertebrae (*Williams et al., 2019*), and we use L2 in these individuals as well for consistency. Thirty-six modern humans, 28 chimpanzees, 26 western gorillas, and 8 orangutans were included. For this analysis, we also utilized a sample of 23 bonobos (*Pan paniscus*) and 7 eastern gorillas (*G. beringei*).

U.W.88–233 is a complete third lumbar vertebra, but it is partially encased in breccia, which obscures some morphologies. The lumbar new vertebrae (U.W.88-232-234) were µCT scanned in partial articulation (*Figure 2*, *Figure 2—figure supplement 1*) at the University of the Witwatersrand using a Nikon Metrology XTH 225/320 LC system. Scan settings were 70 kV, 120 µA, 1 s exposure time, and 3000 projections. Voxel size was 0.049 mm and scans included 2000 voxels. The high-resolution µCT scans were processed to yield virtual 3D models. Each vertebra was segmented using Amira 6.2 (Thermo Fisher Scientific, Waltham, MA). After importing µCT scan slices (TIFF files) and creating a volume stack file (.am), an *Edit New Label Field* module was attached to the stack file. Voxels were selected and assigned to each model separately using the *magic wand* and *brush* tools after verification in all three orthogonal views. A *Generate Surface* module was used to produce a *labels* file (.labels.am) once an individual element was completely selected. A 3D surface model was created from the *labels* file using an unconstrained smoothing setting of 5. Models of each element were then saved as polygon (.ply) files. Using GeoMagic Studio software (3D Systems, Rock Hill, SC), broken portions of U.W.88–233 were refitted and the specimen was reconstructed accordingly. The affected portions of the neural arch were pulled dorsally to refit the fractured portion of the left lamina; additionally, the broken and deflected costal process was refitted. The result is a reconstructed 3D model (*Figure 4*).

Due to crushing of the right SAF, we collected landmarks on the left side of U.W.88–233 and our comparative sample of middle lumbar vertebrae (*Table 1*). Our 3D landmark set consisted of 48 landmarks distributed across the vertebra to reflect the gross morphology (Appendix 1). Landmarks were

collected using the *Landmarks* tool in Amira on the surface model of U.W.88–233 and on 3D models of middle lumbar vertebrae produced using Artec Studio 14 software (Source Graphics, Anaheim, CA).

We used 'geomorph' package version 4.0 (*Adams et al., 2021*) in R version 4.0.2 (*Core Team, 2020*) to carry out 3D GM analyses. The geomorph package was then used to subject the raw landmark data to GPA to correct for position, rotation, and size adjustment. The GPA shape scores were then subjected to PCA using the covariance matrix. We evaluated the effects of centroid size on shape using Procrustes distance-based ANOVA on GPA shape scores as implemented in the geomorph package (*Goodall, 1991*; *Adams et al., 2021*). Specifically, we evaluated the effect of centroid size as a predictor of middle lumbar shape coordinates within each genus by including a genus interaction term (shape~centroid size * genus). Finally, we analyzed two datasets: one on the full set of 48 landmarks in which only complete (reconstructed) fossils (U.W.88–233, Sts 14 c, Shanidar 3) were included, and one on a 37 landmark subset with 11 landmarks on the costal and spinous processes removed so that additional, less well-preserved fossils could be included (A.L. 288-1aa/ak/al, StW 431, Kebara 2).

## Acknowledgements

We thank the University of the Witwatersrand and the Evolutionary Studies Institute, as well as the South African National Centre of Excellence in PalaeoSciences and Bernhard Zipfel and Sifelani Jirah for curating the *A. sediba* material and allowing us access to it and to fossil comparative material in the Phillip V Tobias Fossil Primate and Hominid Laboratory. We are grateful to Kudakwashe Jakata and Kristian Carlson for µCT scanning the *A. sediba* fossils, and Kristian, Morgan Hill, and Erik Mazelis for help processing the µCT scans. We thank the South African Heritage Resource agency for the permits to work at Malapa, and the Nash family for granting access to the site and continued support of research on their reserve, along with the South African Department of Science and Technology, the Gauteng Provincial Government, the Gauteng Department of Agriculture, Conservation and Environment and the Cradle of Humankind Management Authority, the South African National Research Foundation and the African Origins Platform, the National Geographic Society, the Palaeontological Scientific Trust (PAST), and the University of Witwatersrand's Schools of Geosciences and Anatomical Sciences and the Bernard Price Institute for Paleontology for support and facilities, as well as our respective universities. We thank the following individuals for curating and providing access to comparative materials in their care: Mirriam Tawane, Stephany Potze, and Lazarus Kgasi (Ditsong National Museum of Natural History); Brendon Billings and Anja Meyer (Dart Collection, University of the Witwatersrand); Yonas Yilma, Tomas Getachew, Jared Assefa, and Getachew Senishaw (National Museum of Ethiopia and Authority for Research and Conservation of Cultural Heritage); Emma Mbua (National Museums of Kenya); Véronique Laborde, Liliana Huet, Dominique Grimaud-Hervé, and Martin Friess (Musée de l'Homme); Rachel Ives (the Natural History Museum, London); Wim Wendelen and Emmanuel Gilissen (Musée Royal de l'Afrique Centrale); Lyman Jellema and Yohannes Haile-Selassie (Cleveland Museum of Natural History); and Gisselle Garcia, Ashley Hammond, Eileen Westwig, Eleanor Hoeger, Aja Marcato, Brian O'Toole, Marisa Surovy, Sarah Ketelsen, and Neil Duncan (American Museum of Natural History). Bill Kimbel and Chris Stringer facilitated access to fossils, and Erik Trinkaus shared high-quality casts of Kebara 2 and Shanidar 3.

## Additional information

### Funding

| Funder | Grant reference number | Author |
|---|---|---|
| Leakey Foundation | | Scott A Williams |
| Agencia Estatal de Investigación | Museo Nacional de Ciencias Naturales CSIC PID2020-115854GB-I00 | Markus Bastir |

The funders had no role in study design, data collection and interpretation, or the decision to submit the work for publication.

## Author contributions
Scott A Williams, Conceptualization, Data curation, Formal analysis, Funding acquisition, Investigation, Methodology, Project administration, Visualization, Writing – original draft, Writing – review and editing; Thomas Cody Prang, Marc R Meyer, Data curation, Formal analysis, Investigation, Methodology, Visualization, Writing – original draft, Writing – review and editing; Thierra K Nalley, Data curation, Investigation, Methodology, Visualization, Writing – review and editing; Renier Van Der Merwe, Data curation, Writing – original draft, Writing – review and editing; Christopher Yelverton, Kelly R Ostrofsky, Jennifer Eyre, Shahed Nalla, Data curation, Writing – review and editing; Daniel García-Martínez, Data curation, Investigation, Methodology, Writing – review and editing; Gabrielle A Russo, Formal analysis, Methodology, Writing – original draft, Writing – review and editing; Jeffrey Spear, Investigation, Methodology, Writing – review and editing; Mark Grabowski, Markus Bastir, Data curation, Methodology, Writing – review and editing; Peter Schmid, Steven E Churchill, Project administration, Writing – review and editing; Lee R Berger, Conceptualization, Data curation, Funding acquisition, Project administration, Visualization, Writing – review and editing

## Author ORCIDs
Scott A Williams [ID] http://orcid.org/0000-0001-7860-8962
Thomas Cody Prang [ID] http://orcid.org/0000-0003-3032-8309
Marc R Meyer [ID] http://orcid.org/0000-0002-3938-0173
Thierra K Nalley [ID] http://orcid.org/0000-0002-4296-2940
Christopher Yelverton [ID] http://orcid.org/0000-0001-5108-3641
Daniel García-Martínez [ID] http://orcid.org/0000-0001-7518-3866
Gabrielle A Russo [ID] http://orcid.org/0000-0002-2203-1831
Kelly R Ostrofsky [ID] http://orcid.org/0000-0002-7158-546X
Jeffrey Spear [ID] http://orcid.org/0000-0002-0290-7090
Jennifer Eyre [ID] http://orcid.org/0000-0001-6418-6113
Mark Grabowski [ID] http://orcid.org/0000-0001-7045-9472
Shahed Nalla [ID] http://orcid.org/0000-0002-0957-1067
Markus Bastir [ID] http://orcid.org/0000-0002-3141-3401
Lee R Berger [ID] http://orcid.org/0000-0002-0367-7629

## Decision letter and Author response
Decision letter https://doi.org/10.7554/eLife.70447.sa1
Author response https://doi.org/10.7554/eLife.70447.sa2

---

# Additional files

## Supplementary files
• Transparent reporting form

## Data availability
All data generated or analyzed during this study are included in the manuscript and supporting files. Source data files have been provided for Figures 5 and 6 (Figure 5-source data 1, Figure 6-source data 2), and raw XYZ coordinate files for each specimen are available for download on Dryad (https://doi.org/10.5061/dryad.6m905qg0x).

The following dataset was generated:

| Author(s) | Year | Dataset title | Dataset URL | Database and Identifier |
|---|---|---|---|---|
| Williams SA, Thomas P | 2021 | XYZ coordinates of middle lumbar vertebrae - 3D GM analysis for: A nearly complete lower back of Australopithecus sediba | https://doi.org/10.5061/dryad.6m905qg0x | Dryad Digital Repository, 10.5061/dryad.6m905qg0x |

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

## Appendix 1

### Linear and angular measurements

The following measurements were taken on original fossil material or rendered surface models generated from high-resolution µCT scans (see descriptions in Results). Below:

1. Vertebral body superior transverse width (Martin measurement #7; M7): defined in *Bräuer, 1988*, as the superior vertebral body transverse diameter at the most laterally projecting points.
2. Vertebral body superior dorsoventral length (M4): defined in *Bräuer, 1988*, as the superior vertebral body DV diameter measured at the sagittal midline.
3. Vertebral body inferior transverse width (M8): defined in *Bräuer, 1988*, as the inferior vertebral body transverse diameter at the most laterally projecting points.
4. Vertebral body inferior dorsoventral length (M5): defined in *Bräuer, 1988* as the inferior vertebral body DV diameter measured at the sagittal midline.
5. Vertebral body SI ventral height (M1): defined in *Bräuer, 1988*, as the ventral SI height of the vertebral body at the sagittal midline.
6. Vertebral body SI dorsal height (M2): defined in *Bräuer, 1988*, as the dorsal SI height of the vertebral body at the sagittal midline.
7. Vertebral body wedging angle: calculation provided in *Digiovanni et al., 1989*, as [arctangent ((((SI dorsal height-superior DV length)/2)/SI ventral height)*2)].
8. Vertebral foramen dorsoventral length (M10): defined in *Bräuer, 1988*, as DV vertebral foramen diameter measured at the sagittal midline.
9. Vertebral foramen transverse width (M11): defined in *Bräuer, 1988*, as transverse vertebral foramen diameter measured at the roots of the vertebral arch.
10. SI inter-articular facet height: SI inter-articular facet distance, measured from the most superior aspect of the SAF to the most inferior aspect of the IAF on the same side. The right side is measured unless it is broken or pathological, in which case the left side is measured. If the two sides are asymmetrical and one is not pathological, the mean is recorded.
11. Maximum inter-SAF transverse width: maximum (max.) inter-SAF distance, measured from the lateral aspect of one SAF to the lateral aspect of the other.
12. Minimum inter-SAF transverse width: minimum (min.) inter-SAF distance, measured from the medial aspect of one SAF to the medial aspect of the other.
13. Maximum inter-IAF transverse width: maximum (max.) inter- IAF distance, measured from the lateral aspect of one IAF to the lateral aspect of the other.
14. Minimum inter-IAF width: minimum (min.) inter-IAF distance, measured from the medial aspect of one IAF to the medial aspect of the other.
15. SAF SI height: SAF SI diameter, measured at the sagittal midline. The right side is measured unless it is broken or pathological, in which case the left side is measured. If the two sides are asymmetrical and one is not pathological, the mean is recorded.
16. SAF transverse width: SAF transverse diameter, measured from the most medial to the most lateral border of the articular surface. The right side is measured unless it is broken or pathological, in which case the left side is measured. If the two sides are asymmetrical and one is not pathological, the mean is recorded.
17. IAF SI height: IAF SI diameter, measured at the sagittal midline. The right side is measured unless it is broken or pathological, in which case the left side is measured. If the two sides are asymmetrical and one is not pathological, the mean is recorded.
18. IAF transverse width: IAF transverse diameter, measured from the most medial aspect to the most lateral border of the articular surface. The right side is measured unless it is broken or pathological, in which case the left side is measured. If the two sides are asymmetrical and one is not pathological, the mean is recorded.
19. Spinous process angle (M12): defined in *Bräuer, 1988*, as the angle that is formed from the superior surface of the vertebral body and the upper edge of the spinous process. We modify this measurement slightly by measuring the angle along its long axis, which allows for the inclusion of fossils with a damaged or missing superior edge of the spinous process. An angle of 180° is equivalent to a spinous process with a long axis parallel to the superior surface of the vertebral body (i.e., horizontal or neutral in orientation).

20. Spinous process length (M13): defined in *Bräuer, 1988*, as the distance from the top edge of the vertebral arch to the most dorsal tip of the spinous process.
21. Spinous process terminal transverse width: transverse breadth of the dorsal tip of the spinous process, measured at its maximum dimension.
22. Spinous process terminal SI height: SI diameter of the dorsal tip of the spinous process, measured at the mediolateral midline of the spinous process.
23. Costal (transverse) process SI base height: SI diameter of the costal process at its medial origin from the pedicle and/or vertebral body.
24. Costal process dorsoventral angle: the dorsoventral angle that is formed from the sagittal midplane of the vertebra to the long axis of the costal process, along the middle from its base to its tip. The right side is measured unless it is broken or pathological, in which case the left side is measured. If the two sides are asymmetrical and one is not pathological, the mean is recorded.
25. Costal process length: the distance from the internal edge of the vertebral foramen at its closest point to the base of the costal process to the tip of the costal process. The right side is measured unless it is broken or pathological, in which case the left side is measured. If the two sides are asymmetrical and one is not pathological, the mean is recorded.
26. SAF orientation: the angle formed between the sagittal midplane of the vertebra to the medial and lateral edges of the SAF. The right side is measured unless it is broken or pathological, in which case the left side is measured. If the two sides are asymmetrical and one is not pathological, the mean is recorded.
27. Pedicle SI height: SI diameter of the pedicle, measured at its midpoint. The right side is measured unless it is broken or pathological, in which case the left side is measured. If the two sides are asymmetrical and one is not pathological, the mean is recorded.
28. Pedicle transverse width: transverse breadth of the pedicle, measured at its midpoint. The right side is measured unless it is broken or pathological, in which case the left side is measured. If the two sides are asymmetrical and one is not pathological, the mean is recorded.
29. Pedicle dorsoventral length: DV diameter of the pedicle, measured anterior from its junction with the superior articular process to its junction with the dorsal edge of the vertebral body. The right side is measured unless it is broken or pathological, in which case the left side is measured. If the two sides are asymmetrical and one is not pathological, the mean is recorded.
30. Lamina SI height: SI dimension of the lamina, measured on one side between the spinous process and the SAF and the IAF. The right side is measured unless it is broken or pathological, in which case the left side is measured. If the two sides are asymmetrical and one is not pathological, the mean is recorded.
31. Lamina transverse width: transverse dimension of the lamina, measured at its minimum breadth across the *pars interarticularis*.

## List of 3D landmarks

The following 3D landmarks were collected on middle lumbar vertebrae using AMIRA:

1. Superior vertebral body – ventral transverse midline
2. Superior vertebral body – central transverse midline
3. Superior vertebral body – dorsal transverse midline
4. Superior vertebral body – lateral sagittal midline
5. Superior vertebral body – ventro-lateral point
6. Superior vertebral body – dorso-lateral point (at ventral pedicle base)
7. Pedicle – superior midpoint
8. Pedicle – superior dorsal point (at ventral base of prezygapophysis)
9. Pedicle – medial midpoint
10. Pedicle – lateral midpoint
11. Pedicle – inferior midpoint
12. *Pars interarticularis* – dorsal midpoint
13. *Pars interarticularis* – ventral midpoint
14. Inferior vertebral body – ventral transverse midline
15. Inferior vertebral body – central transverse midline

16. Inferior vertebral body – dorsal transverse midline
17. Inferior vertebral body – lateral sagittal midline
18. Inferior vertebral body - ventro-lateral point
19. Inferior vertebral body – dorso-lateral point
20. Costal process – superior medial point (based of costal process at pedicle)
21. Costal process – inferior medial point (based of costal process at pedicle)
22. Costal process – superior mediolateral midpoint
23. Costal process – inferior mediolateral midpoint
24. Costal process – superior lateral point
25. Costal process – inferior lateral point
26. SAF – cranial-most point
27. Mammillary process – dorsal-most extension
28. SAF – midpoint
29. SAF – caudal-most point
30. Spinous process – superior ventral point (at spinous process base)
31. Spinous process – superior sagittal midpoint
32. Spinous process – superior dorsal point (tip of spinous process)
33. Spinous process – inferior dorsal point (tip of spinous process)
34. Spinous process – inferior sagittal midpoint
35. Spinous process – inferior ventral point (at spinous process base)
36. Lamina – inferior midpoint
37. IAF – cranial-most point
38. IAF – midpoint
39. IAF – caudal-most point
40. IAF – medial-most point
41. IAF – lateral-most point
42. SAF – medial-most point
43. SAF – lateral-most point
44. Vertebral body – ventral midpoint
45. Vertebral body – lateral midpoint
46. Costal process – ventral extension of costal process base at its midpoint
47. Spinous process – lateral-most extension of the spinous process tip
48. Lamina – superior midpoint

