## [Decision Letter]

**Acceptance summary:**

Williams et al. present new fossil remains from the lower back of one of the specimens of Australopithecus sediba, the Malapa Hominin 2 (MH2), so that now almost the complete lumbar spine of this important australopithecine specimen is known. This paper is a very valuable contribution to paleoanthropology especially to those who study the evolution of the vertebral column in hominins.

**Decision letter after peer review:**

Thank you for submitting your article "New fossils of Australopithecus sediba reveal a nearly complete lower back" for consideration by *eLife*. Your article has been reviewed by 2 peer reviewers, and the evaluation has been overseen by a Reviewing Editor and Detlef Weigel as the Senior Editor. The following individual involved in review of your submission has agreed to reveal their identity: Martin Hauesler (Reviewer #2).

Essential revisions:

Considering that your conclusions regarding human-like bipedalism are not completely supported by your results, we ask that you state your interpretations related to the locomotor behavior of this specimen in a more cautious manner.

*Reviewer #1 (Recommendations for the authors):*

This study aims to present new fossil remains from the lower back of one of the specimens of A. sediba (MH2). The authors identified portions of four lumbar vertebrae (L1-L4). The L3 is near complete, and they focussed most of the analyses on this vertebra.

Overall, this is a very interesting and valuable contribution to paleoanthropology. However, I have some major concerns that should be addressed before acceptance. In general, the methods need to be explained in more detail, and some of the analyses need to be revised. Also, another concern is that in some parts of the paper, the authors try to demonstrate that the lower back of MH2 is human-like instead of describing more neutrally these very valuable fossils. In this sense, I think this is where the authors try to do more than it is directly set up to do. If the authors want to test if the lower back of MH2 shows more or less human-like traits that might suggest a kind of bipedal locomotion, the inclusion of hypotheses to test are necessary. I will be happy to review any other versions.

Specific comments:

Line 47: I agree based on the morphology of the fossils, that A. sediba used its lower back in a kind of bipedalism. However, the mosaic of features shown in the lower back tells us that we should be cautious to affirm that was a “human-like bipedalism”. The authors should find another way to define it, human-like bipedalism (which is an obligated or complete bipedalism) is not demonstrated here.

Lines 84-103: A figure representing the site, road, the site, etc, indicating where the fossils were found, would be very helpful to understand the context. This is necessary when new fossils are presented.

Line 110: Please, specify here the hypotheses you test in this work.

Line 116: As I comment (see below), these measurements could have been used to test/compare MH2 with other hominin fossils and extant hominids as a complementary analysis to GMM. However, I appreciate you included the measurements and the 3D digital models.

Line 141: Please clarify why do identify these vertebrae as L2 and L3. Some arguments are needed to assign anatomical identification to new fossils. If is just because they refit with the previously known vertebrae, say it, if you have other anatomical arguments, describe them.

Line 189: You have used the 95 % of confidence for other analyses (see Line 244), but here you have included the 100 % of the modern human variability. Also, morphological differences between hominins and great apes are very large and therefore, differences between modern humans and fossils can be diluted. I would suggest repeating this PCA including only modern humans and the three fossil remains with a 95% confidence ellipse. I am not saying to remove these plots from the manuscript, but an extra analysis to zoom on what is the most interesting comparative. We do not doubt that MH2 is going to plot closer to modern humans than to orangutans.

Lines 205-206: I don't agree with the statement referring to the vertebral body shape. I think the second analysis, that excluding the spinous process and transverse process, is a better proxy for analyzing differences or similarities of the vertebral shape. The results from that analysis (Lines 213-215) contradict this affirmation and suggest that the vertebral shape of U.W.88-233 (and that of Sts 14) is not within the modern human's variability. Said that I appreciate the effort of including more fossil remains in the sample.

Line 219: There exist specific functions inside geomorph package to test for allometry. This ANOVA is not enough to affirm that all the groups present the same (parallel) allometric vector. If the authors want to affirm this, please, test it properly.

Line 275: I would like to know which hypotheses are you testing here, and discuss the results of these tests in this section.

Line 276: These three characteristics are a demonstration of modern human-like morphology rather than bipedal primary adaptations. For example, Neandertals were also bipeds and did not show that degree of lumbar lordosis. I understand that a human-like lumbar morphology might suggest a similar locomotor behavior, but state that “demonstrates” is going too far.

Line 352: Analyses based on linear or angular variables, are not 2D analyses. Indeed, are univariate variables and therefore unidimensional analyses.

Line 354: High-quality what? If is a scanned 3D model, please specify the technical characteristics of it (resolution, etc.).

Lines 354-357: Please, this is the “Materials” section, be specific with the fossil remains used in this study as you do with the comparative sample. Also, indicate whether the fossil remains are a second, third, fourth, or fifth lumbar vertebra.

Line 357: Why varied among analyses?

Line 370: What do you mean by "we only include well sampled chimpanzees and western gorillas from the comparative sample”? What about the other analyses, aren't they well sampled?

Line 373: Specify how (software, function) did you perform this resampling.

Line 375: Clarify why you used different vertebral elements to measure this distance.

Line 383: I guess I know why, but please, explain why you use L2 as a middle lumbar vertebra in great apes and not L3.

Lines 384-386: For the 3D analyses, did you also select exclusively those individuals with four lumbar vertebrae? Eastern gorillas have a vertebral formula of 7:13:3:6, that is, they usually have three lumbar vertebrae. Also, in Line 386, add graueri to Gorilla beringei, otherwise, that could also refer to mountain gorillas (Gorilla beringei beringei).

Line 387: Be specific with the anatomical determination of the vertebra, “U.W.88-233 is a complete third lumbar vertebra”.

Line 388: Lumbar block is not appropriated. It could be the block including the fossil lumbar vertebrae or something similar.

409: I don't think a PCA is a good method to detect outliers. Your sample includes very diverse species, and an outlier may not be detected in such a huge variability. There are other, more specific, and appropriate methods to detect outliers. Please, modify this, and apply other functions such as plotOutliers from geomorph, or find.outliers from Morpho (or any other similar to these). The latter also includes a visual interface to detect possible mistakes in landmarking that are difficult to detect simply by plotting the sample looking for outliers.

412: Do you also use geomorph to perform the PCA? I am a bit confused with the plots you obtained. Also, I never saw those wireframes to visualize shape changes. Clarify where did you get them from. In addition, why don't you use those 3D models you have to obtain a more clear and visual representation of the shape's extreme morphologies? In the list of co-authors, some researchers usually represent shape variation showing modified 3D models.

Line 412: Here I miss the goal of the regression analysis. This goes together with the lack of hypotheses to test, which could justify the methodological procedure of this work. Also, there is a very important lack of information about how this analysis was carried out. In the Results section, there is more information about the procedure than in the section “Methods” itself. Please, explain here step by step how did you perform this analysis. Also, in the Results section you mention that you grouped the species into genera pooling hominins together, but to perform a regression analysis including different species pooled into genera, previously you should remove the within-group variation.

Line 415: In the same vein, you mention you carried out two analyses. But these are not analyses themselves. You used a different number of landmark coordinates to perform different analyses, which in addition you do not mention here. Please rewrite this paragraph and explain what you did.

Line 595: Please describe the first “block” either from caudal to cranial or vice versa, but following an anatomical order. You start by the sacrum, then L5, then L4, L5 again, and finally S1.

In the second block you follow the cranial-caudal direction for describing it, please, be consistent (same for the first block).

Line 628: This is my concern about MH2 human-like vertebral body morphology. Once you remove the spinous process and the transverse processes, basically what we have is the vertebral body and the articular facets. The results indicate that australopiths are outside the modern humans' range of variation.

Lines 637-642: All this information should be in a table. It is difficult to compare and follow all the information you show here.

Line 645: Regarding the column with the values of gorillas and chimpanzees, clarify what Table are you referring to (Excel Table with the raw data for Figure 6 I guess).

Line 655: Thanks for providing all this information. This is just my point of view, and I will leave it in the hands of the AE, but I think a simple non-parametric analysis (e.g. Mann-Whitney u test) comparing these traits between extinct and extant species would have been very useful to assess specific significant differences (or similarities) between MH2 lumbar vertebrae and those of other australopiths, Neanderthals, modern humans, and great apes. The results could have (or not) demonstrate in specific traits (e.g. vertebral body) the similarities between modern humans and MH2 you propose as demonstrated by other methods (and I have my doubts after your results).

Line 856: Describe the views as in the other A, B, and C figures. Also, provide the number of the landmarks, if not all because of the lack of space, at least most of them.

Line 859: These analyses are not explained in the Methods section. Neither the obtained results. Include or remove them.

Line 860: “between U.W. 88-233 and extant and fossil middle lumbar vertebrae” Here only some of the extant species included in the sample are analyzed. Why? Specify which species you included.

Line 863: Could you please explain what is this within (Homo, Pan) (mean Procrustes distance?), or better said, the purpose of comparing that mean? Also, in this Figure 3 (B), why there is not lateral view as in Figure 3 (C)?

Line 868: Here it would be interesting to comment that both australopiths plot within the chimpanzees' range of variation, and the Neandertal within (or in the limit) of modern humans.

*Reviewer #2 (Recommendations for the authors):*

Please always use the correct anatomical terms according to the current version of the Terminologia anatomica. The "lumbar transverse process" is an erroneous, antiquated and obsolete term that needs to be replaced by "costal process" throughout the paper. See also Strzelec, B., Chmielewski, P.P., Gworys, B., 2017. The Terminologia Anatomica matters: examples from didactic, scientific, and clinical practice. Folia Morphologica 76, 340-347.

Likewise, the term "spinal canal" designates the canal that is formed by the vertebral foramina of several subsequent vertebrae for the spinal cord, but in the individual vertebra it would be termed "vertebral foramen".

A more stylistic issue is the use of the colloquial term "australopith". This word represents a sloppy mutilation of "pithekos", the Greek word for "ape". The syllable "pith" in australopith is nonsensical. Some people say they prefer this slang because they claim that the "australopithecine" would imply monophyly of the genera Australopithecus and Paranthropus. This is, however, not correct. Australopithecines is not only the English term for the former subfamily "Australopithecinae", but also for the subtribe "Australopithecina" of modern systematics. Therefore, I strongly urge the authors to replace "australopiths" with "australopithecines"

The authors compare the morphology of the newly described, complete third lumbar vertebra U.W. 88-233 of A. sediba to the "middle lumbar vertebrae" of modern humans, chimpanzees and gorillas. It is for me, however, unclear, what they exactly mean with "middle lumbar vertebra" because australopithecines and modern humans possess on average 5 lumbar vertebrae and chimpanzees and gorillas generally have (3-) 4. Could the authors therefore shortly discuss the issue of homology and how the outcome would be affected if U.W. 88-233 and the human L3 would be compared to L2 vs. L3 in the great apes.

L213 "Humans and great apes separate along PC1, which is largely explained by vertebral body height": isn't it rather vertebral wedging?

L214 "U.W.88-233 and Sts 14 fall intermediately between modern humans and great apes along PC1": This should be changed to say that this applies to all australopithecines.

L237 "a ventral curvature of the lumbar spine": a lordotic curve cannot be described as a ventral curvature-it is a ventrally convex curvature.

L241 "sum L2-L5 wedging" is awkward. Consider something like "total L2-L5 wedging angle"

L248 "see Figure 6 caption": Unfortunately, I don't see the data for chimpanzees in in Figure 6., and also the caption to Figure 6 does not provide more relevant information. On the other hand, I don't think it is a good option to use figure captions to present additional information that is otherwise not contained within the manuscript.

L249 "Patterns of change across lumbar levels demonstrate that MH2's vertebrae transition from ventral (kyphotic) wedging at the L2 (most similar to male modern humans) and L3 (most similar to female modern humans) levels to dorsal (lordotic) wedging at L4. At the L4 level MH2 is most similar to female modern humans and female…": In my opinion, inferences for a greater similarity of the vertebral wedging patterns of fossil hominins to a specific modern humans sex is overinterpreting the data. Thus, the considerable variability of these wedging patterns within modern humans needs to be taken into account. In fact, the data presented in Figure 6 suggest that both males and females transition from ventral to dorsal wedging between L3 and L4. This also relates to the quite misleading use of the 95% confidence intervals of the means in Figure 6 rather than the 95% confidence intervals of the sample data (see my comment to Figure 6).

L270 Maybe the authors can briefly discuss the functional significance of the pyramidal configuration for lumbar lordosis and cite the corresponding literature. It would also be good to discuss Oreopithecus in this context, for which a similar pyramidal configuration has been reported (Köhler, M., Moyà-Solà, S., 1997. Ape-like or hominid-like – the positional behavior of Oreopithecus bambolii reconsidered. Proceedings of the National Academy of Sciences of the United States of America 94, 11747-11750.)

L288 "ample surface area": isn't the length of the lever arm of the costal processes more relevant than the relatively moderate increase in surface area (if at all) compared to that of modern humans? And why do australopithecines need more powerful trunk musculature than modern humans? I also don't understand, why a cranial orientation of the costal processes should increase the "moment arms and torque generation capabilities of psoas major and quadratus lumborum". Isn't the lever arm only related to its length perpendicular to the force vector? I therefore think that the cranial orientation of the costal process must have another explanation.

L290 There is no "middle lumbar fascia", but the anterior and middle layers of the thoracolumbar fascia insert on the costal processes. On the other hand, the m. obliquus externus abdominis has generally no contact to the thoracolumbar fascia in modern humans as far as I know.

L295 Please rephrase. Muscles cannot "support" the pelvis. Support means "bear all or part of the weight of something", but as muscles cannot counteract compressive forces they cannot bear weight. Moreover, the support of a structure must be below the structure itself.

L299 Please also see Tardieu et al. (2017. How the pelvis and vertebral column became a functional unit in human evolution during the transition from occasional to permanent bipedalism? Anatomical Record 300, 912-931) who show that the pelvic incidence of alternative pelvic reconstructions of MH2 falls well within the range of modern humans.

L303 The analysis of La Chapelle-aux-Saints demonstrated that Neanderthals possessed a well-developed lordosis similar to modern humans, see Haeusler et al. (2019. Morphology, pathology and the vertebral posture of the La Chapelle-aux-Saints Neandertal. Proceedings of the National Academy of Sciences of the United States of America 116, 4923-4927). Moreover, the reported low pelvic incidence of SH1 is probably misleading because it is due to a lumbosacral transitional anomaly, see Haeusler, M., 2019. Spinal pathologies in fossil hominins, in: Been et al.. (Eds.), Spinal Evolution: morphology, function, and pathology of the spine in hominoid evolution. Springer, Cham, pp. 213-245.

L308 Please see my other comments. I don't think it is possible to differentiate the sex based on the wedging values.

L309 This is not true. Schiess et al. (2014. Revisiting scoliosis in the KNM-WT 15000 Homo erectus skeleton. Journal of Human Evolution 67, 48-59.) demonstrated that KNM-WT 15000 (Homo erectus) possesses an even slightly stronger dorsally wedged L5 vertebra than MH2. Nevertheless, it was still within the 95% range of variation of modern humans (see their Table 4 and Figure 5D)

L337 Please mention also that its presence has been claimed for Oreopithecus (see above)

L354 „high-quality" seems to miss "scans" or something similar. It would be useful if the authors could indicate where they come from. At least the scans of Shanidar3 that are distributed by the Smithsonian can nowadays hardly be called "high quality". Please also note that the Shanidar vertebrae are extensively reconstructed.

L357 I'm not aware that any of the fossils used in the present study are curated at the NHM London. Please check.

L365 I wasn't aware that also modern human skeletons are curated at the Musée Royale de l'Afrique Centrale. Please check.

L368 Please note that this is a basic trigonometry approach that has been used by DiGiovanni et al., and I therefore recommend to say this rather than citing DiGiovanni et al. There is also no fancy "wedging angle equation" necessary, but the simple use of the tangent. On the other hand, this equation is not provided in Table 1 as the text suggests, so it might be good to rephrase this sentence.

L372 Why this? What is needed here is the 95% range of variation or more precisely, the 3rd to 97th inter-percentile range.

Figure 6 Please show the 95% confidence intervals of the sample data, not the 95% confidence intervals of the means for modern humans males and females. This misleading and does not acknowledge the considerable range variation that is typical for these wedging data (see e.g., Figure 5D in Schiess et al. 2014).

L637-642 Please consider to report these values in a table rather than in the figure caption

Figure 7 Please use the correct scientific abbreviations! On the other hand I don't think it is still fine to use H. neanderthalensis for the Neanderthals since genetic data clearly demonstrate interbreeding with modern humans

[Editors' note: further revisions were suggested prior to acceptance, as described below.]

Thank you for resubmitting your work entitled "New fossils of Australopithecus sediba reveal a nearly complete lower back" for further consideration by *eLife*. Your revised article has been evaluated by 2 reviewers and the evaluation has been overseen by a Reviewing Editor and Detlef Weigel as the Senior Editor.

The manuscript has been improved but there are few remaining issues that need to be addressed.

One of the issues concerns the wedging angles. The authors compare the wedging angles of the MH2 lumbar vertebrae with the 95% confidence intervals of the mean female and the mean male modern human wedging angles. Generally, these 95% confidence intervals about the means are useful if the purpose is to compare two samples and to explore whether their means are different. However, the mean values are unknown for A. sediba as a species (or any other fossil hominin species). Only those of a single A. sediba individual, MH2, are known, and a comparison of the means of A. sediba with those of modern humans is therefore unwarranted. We can only compare the MH2 specimen with the modern human sample by hypothesizing that there is no difference between A. sediba and *H. sapiens* for this trait in terms of the distribution of the data points of the entire species (=H1a). This means that individual data points will of course differ from the mean and form a Gaussian distribution about the hypothetical mean value. Under this assumption, the data point for A. sediba would therefore fall with a 95% probability within a certain range. This is represented by the 95% confidence interval of the sample (also known as the 95% prediction interval). Thus, the 95% prediction interval represents the range of values that likely contains the value of a new observation given the distribution of the comparative sample. This 95% prediction interval can be approximated by 2 standard deviations (more precisely, it would be 1.96×standard deviations), and the authors now also show this range in their Figure 5, but unfortunately, they do not use this interval further and do not discuss it in the text, but inappropriately base their discussion and conclusion on the 95% confidence intervals about the mean.

*Reviewer #1 (Recommendations for the authors):*

I am satisfied that the authors have addressed most of my concerns, especially those regarding the interpretation of their results. I am also glad about the inclusion of the hypotheses, which I think help to better understand and follow the purpose of this work. The authors have explained in greater detail some methodological aspects that needed some clarification. All in this, the discussion is much more solid and coherent with the results than the previous version. In general, the authors have done a great good job. As I assessed in my first revision, these fossils are a great contribution to paleoanthropology, especially to the study of the evolution of the vertebral column in hominins. Thus, I recommend the publication of this manuscript after correcting a few details.

First, I have some doubts about one aspect of the rebuttal document:

The only point I think was not clarified in my previous revision referred to Supplementary Figure 3 (from the first version). I wrote: "Line 859: These analyses are not explained in the Methods section. Neither the obtained results. Include or remove them.".

The authors have clarified that: "It was in fact referenced in both the Methods (penultimate paragraph) and in the Discussion (p. 10):. In the main text, we wrote, "We plotted standardized shape scores derived from a multivariate regression of shape on centroid size against centroid size to visualize shape changes (Drake and Klingenberg, 2008) (Supplementary Figure 4)."

But this refers to Supplementary figure 4 (Line 868), and not to Supplementary figure 3, which corresponds with Line 859. I appreciate they clarified this part, but the analyses and results from this Supplementary Figure 3 are still not explained in the manuscript. This figure reads "Procrustes distances and mean differences…" but neither a reference to Procrustes distances nor to mean differences appear in the entire manuscript apart from this footnote. In the rebuttal to my concern about Line 863, which also refers to this figure, they explain in detail what this figure means, and I appreciate it, but this should also be in the manuscript.

*Reviewer #2 (Recommendations for the authors):*

The revised manuscript has improved in many aspects. Particularly, it is now more than a simple exploratory study, having a greater focus on hypothesis testing. However, the wrong use of statistics in the analysis and hypothesis testing of the wedging angles still represents a major issue that needs to be addressed.

Thus, the first hypothesis (H1a) of the current study is that there is no difference between the wedging angles of MH2 and modern humans, and another hypothesis, which I call H1b, is that they are distinct from extant great apes. To test hypothesis H1a, the authors compare the wedging angles of the MH2 lumbar vertebrae with the 95% confidence intervals of the mean female and the mean male modern human wedging angles. Generally, the 95% confidence intervals about the means are useful if the purpose is to compare two samples and to explore whether their means are different. However, the distribution of the wedging angles and thus their mean values are unknown for A. sediba (or any other fossil hominin species). Only those of a single A. sediba individual, MH2, are known, and a comparison of the means of A. sediba with those of modern humans is therefore not possible. We can only compare the MH2 specimen with the modern human sample by hypothesizing that there is no difference between A. sediba and *H. sapiens* for this trait (=H1a). Under this assumption, the data point for A.sediba would therefore fall with a 95% probability within a certain range. This is represented by the 95% confidence interval of the sample (also known as the 95% prediction interval). Thus, the 95% prediction interval represents the range of values that likely contains the value of a new observation given the distribution of the comparative sample. This 95% prediction interval can be approximated by 2 standard deviations (more precisely, it would be 1.96×standard deviations), and the authors now also show this range in their Figure 5, but unfortunately they don't use this interval further and don't discuss it within the text.

In fact, Figure 5 shows that the wedging angles of all lumbar vertebrae of MH2 fall within the 95% prediction intervals of both modern human males and females. The same is true for all other analysed fossil hominins, except for Shanidar 3 and Kebara 2, whose wedging angles of L2 fall only within the male range of the current sample. Because we don't know the 95% confidence intervals about the means of the A. sediba wedging angles (or those of A. africanus, etc.), it is irrelevant whether MH2 (or Sts 14 or StW 431) lies closer to the female or the male means of modern humans for some vertebrae, as they are only some individuals. The corresponding sections in the text (L316-319 and L350-366) should therefore be rephrased accordingly. Likewise, the right side of Figure 5 needs to be adapted to show the 95% prediction intervals rather the 95% confidence intervals of the means.

Hypothesis H1b (that the wedging angles of MH2 are distinct from extant great apes) is only marginally addressed as far as I can see. Thus, wedging angles are only reported in Table 2 for chimpanzees (and thus only for one of three great ape genera). Nevertheless, it seems that the wedging angles of vertebrae L2-L4 of MH2 are well within the 95% prediction intervals for chimpanzees (as approximated by the means {plus minus} 2 SD). Does this therefore mean that lumbar lordosis of MH2 or other fossil hominins cannot statistically be differentiated from that of chimpanzees? Can the authors expand on this? It also would be helpful if the means and the 95% prediction intervals for chimpanzees (and if possible gorillas and orangutans) are included in figure 5 (or in an additional figure).

Regarding my suggestion to include KNM-WT 15000 into the study, I agree with the authors that this is not so easy for the 3D GM analyses due to its subadult age. However, I still maintain that the addition of KNM-WT 15000 would be fundamental to the interpretation of the wedging angles as it shows that the strong lordotic wedging of L5 is not exceptional in MH2 and Kebara 2 (see Schiess et al. 2014). The subadult age of KNM-WT 15000 explains of course the missing vertebral ring apophyses, but this does not affect the wedging angles of the vertebral bodies since the ring apophyses are flat.

---

## [Author Response]

Reviewer #1 (Recommendations for the authors):This study aims to present new fossil remains from the lower back of one of the specimens of A. sediba (MH2). The authors identified portions of four lumbar vertebrae (L1-L4). The L3 is near complete, and they focussed most of the analyses on this vertebra.Overall, this is a very interesting and valuable contribution to paleoanthropology. However, I have some major concerns that should be addressed before acceptance. In general, the methods need to be explained in more detail, and some of the analyses need to be revised. Also, another concern is that in some parts of the paper, the authors try to demonstrate that the lower back of MH2 is human-like instead of describing more neutrally these very valuable fossils. In this sense, I think this is where the authors try to do more than it is directly set up to do. If the authors want to test if the lower back of MH2 shows more or less human-like traits that might suggest a kind of bipedal locomotion, the inclusion of hypotheses to test are necessary. I will be happy to review any other versions.

We greatly appreciate the reviewer’s time in providing detailed comments and lending their time to help improve our manuscript. We think their requests are reasonable (although we think poor word choice on our part is to blame rather than our desire to “try” to make MH2 look more human-like, and we have made changes where appropriate), and we have explained the methods in more detail and revised the analyses and presentation of results as requested.

Specific comments:Line 47: I agree based on the morphology of the fossils, that A. sediba used its lower back in a kind of bipedalism. However, the mosaic of features shown in the lower back tells us that we should be cautious to affirm that was a “human-like bipedalism”. The authors should find another way to define it, human-like bipedalism (which is an obligated or complete bipedalism) is not demonstrated here.

We understand the point the reviewer makes here and think our use of “human-like” was misunderstood. We clarify that we do not think MH2 or any early hominin engaged in modern human-like bipedalism. Instead, we were mainly referring to “human-like” bipedalism to contrast with ape-like bipedalism. Our comparisons of MH2 to modern humans are done because we are the only extant hominins for which we can gather large samples to compare to fossil hominins. We fully agree that MH2 and other early hominins were not fully modern human-like in bipedalism; in fact, we suspect that many modern human adaptations have to do with endurance walking and even running, which would be absent in tree-climbing early hominins like *A. sediba*.

Lines 84-103: A figure representing the site, road, the site, etc, indicating where the fossils were found, would be very helpful to understand the context. This is necessary when new fossils are presented.

We have now included a new supplementary figure showing the location of the mining road at Malapa.

Line 110: Please, specify here the hypotheses you test in this work.

We have now added hypotheses throughout the manuscript. Thank you for this suggestion.

Line 116: As I comment (see below), these measurements could have been used to test/compare MH2 with other hominin fossils and extant hominids as a complementary analysis to GMM. However, I appreciate you included the measurements and the 3D digital models.

We appreciate this suggestion and address it below.

Line 141: Please clarify why do identify these vertebrae as L2 and L3. Some arguments are needed to assign anatomical identification to new fossils. If is just because they refit with the previously known vertebrae, say it, if you have other anatomical arguments, describe them.

It is based in part of their morphology, but largely on the fact that they refit with known elements, the penultimate lumbar vertebra of MH2, which is preserved in articulation with the ultimate lumbar vertebra and the sacrum. In other words, the morphology of the new vertebrae does not contradict their apparent numeration and association with previously known lower lumbar vertebrae. We have made this more clear in the manuscript.

Line 189: You have used the 95 % of confidence for other analyses (see Line 244), but here you have included the 100 % of the modern human variability. Also, morphological differences between hominins and great apes are very large and therefore, differences between modern humans and fossils can be diluted. I would suggest repeating this PCA including only modern humans and the three fossil remains with a 95% confidence ellipse. I am not saying to remove these plots from the manuscript, but an extra analysis to zoom on what is the most interesting comparative. We do not doubt that MH2 is going to plot closer to modern humans than to orangutans.

Thank you for the suggestion. We repeated the 3D GM analysis using only humans with the ‘geomorph’ package version 4.0 in R version 4.0.2. The patterns observed along the first two principal components are very similar to the patterns we detected along PC1 and PC4 in our original analysis (Figure 7c; Figure 7—figure supplement 2). You will see that figure now includes 95% CIs of the human data. We have contextualized this with our functional interpretation of costal processes.

Lines 205-206: I don't agree with the statement referring to the vertebral body shape. I think the second analysis, that excluding the spinous process and transverse process, is a better proxy for analyzing differences or similarities of the vertebral shape. The results from that analysis (Lines 213-215) contradict this affirmation and suggest that the vertebral shape of U.W.88-233 (and that of Sts 14) is not within the modern human's variability. Said that I appreciate the effort of including more fossil remains in the sample.

The reviewer is correct that the spinous and costal processes are driving a lot of the variation in the first PCA and we should focus on the second PCA (without those processes) for vertebral body differences. We have modified the text to reflect the points the reviewer has made.

Line 219: There exist specific functions inside geomorph package to test for allometry. This ANOVA is not enough to affirm that all the groups present the same (parallel) allometric vector. If the authors want to affirm this, please, test it properly.

In general, analysis of variance (ANOVA) is an extension of ordinary least-squares (OLS) regression, but it enables one to examine and partition variance in a response variable in a more intuitive way given that we can examine the extent to which predictor variables (e.g., centroid size) “soak up” variance in a response variable (i.e., lumbar vertebra shape) by examining the *F* statistic, mean square, sum of squares, and residual sum of squares. Both ANOVA and OLS fall under the umbrella of linear modeling.

The only way to test for the effects of centroid size on shape (i.e., “allometry”) in the geomorph package is through the use of the procD.lm function (which can be expanded to include a phylogenetic tree as a variance-covariance matrix in the error term of the linear model in a large comparative study using the function ‘procD.pgls’). According to the geomorph package.PDF: “Prior to geomorph 3.0.0, the function, plotAllometry, was used to perform linear regression of shape variables and size, and produce plots to visualize shape allometries. […] This function coalesces a few plotting options found in other functions, as a wrapper, for the purpose of retaining the plot.procD.allometry options in one place.”

We used the function ‘procD.lm’ in geomorph to evaluate the effect of centroid size on lumbar shape with a genus interaction (i.e., testing whether the estimated effect of centroid size on lumbar shape differed by genus). This function uses an approach that is known as a Procrustes ANOVA (Goodall, 1991), which is equivalent to distance-based ANOVA designs (Anderson, 2001). Another way to interpret the inclusion of the interaction term into the model is that in a simpler univariate model (i.e., response variable ~ predictor variable), the slope and intercept would be estimated separately for each genus. As reported in the manuscript, “The results show significant effects of centroid size (F = 9.83; *p* < 0.001), genus (with hominins pooled; F = 27.7; *p* < 0.001), and an interaction between genus and centroid size (F = 1.48; *p* = 0.01), implying unique shape allometries within genera (Table 4).” In other words, we found that centroid size predicts shape changes in lumbar vertebrae across each genus, and that the way in which shape changes according to changes in centroid size differs across each group. We did not find support for the hypothesis that the “allometric” changes are the same (which *could* be, but do not necessarily have to be, represented by parallel “allometric vectors”).

Line 275: I would like to know which hypotheses are you testing here, and discuss the results of these tests in this section.

We have now added hypotheses throughout the manuscript. Thank you for this suggestion.

Line 276: These three characteristics are a demonstration of modern human-like morphology rather than bipedal primary adaptations. For example, Neandertals were also bipeds and did not show that degree of lumbar lordosis. I understand that a human-like lumbar morphology might suggest a similar locomotor behavior, but state that “demonstrates” is going too far.

We understand the reviewer’s points here and have modified how we talk about this throughout.

Line 352: Analyses based on linear or angular variables, are not 2D analyses. Indeed, are univariate variables and therefore unidimensional analyses.

We have changed the subtitle.

Line 354: High-quality what? If is a scanned 3D model, please specify the technical characteristics of it (resolution, etc.).

“Casts” was accidentally dropped. Casts of Kebara 2 and Shanidar 3 were borrowed from Erik Trinkaus, as mentioned in the acknowledgments. We have added “casts” back in. Thank you.

Lines 354-357: Please, this is the “Materials” section, be specific with the fossil remains used in this study as you do with the comparative sample. Also, indicate whether the fossil remains are a second, third, fourth, or fifth lumbar vertebra.

We now mention all fossil specimens included in the study by museum, specimen number, and vertebral level.

Line 357: Why varied among analyses?

We have dropped this misleading phrase that we used to indicate that not all chimpanzees, for example, were included in all of the analyses. Those with three lumbar vertebrae, for example, were excluded from the vertebral wedging analyses. This is already indicated in the manuscript elsewhere.

Line 370: What do you mean by “we only include well sampled chimpanzees and western gorillas from the comparative sample”? What about the other analyses, aren't they well sampled?

We are emphasizing that they are well-sampled taxa. Other taxa are not well-sampled for some analyses. For example, we did not collect some caliper measurements on eastern gorillas, bonobos, and orangutans, although we do have other measurements and some 3D models of select lumbar vertebrae.

Line 373: Specify how (software, function) did you perform this resampling.

We used PAST 4 to run Bootstrap. We now list PAST 4 version 4.06b and cite Hammer et al. (2001).

Line 375: Clarify why you used different vertebral elements to measure this distance.

It is the coverage of levels that needs to be standardized. Great apes have fewer lumbar vertebrae, so, in humans L2-L5 is used (equivalent to L1-L4 in great apes). There is also a practical reason – L1 is largely not preserved in MH2.

Line 383: I guess I know why, but please, explain why you use L2 as a middle lumbar vertebra in great apes and not L3.

In humans, L3 is obviously the middle of a 5-element lumbar column. L2 is the middle lumbar for many chimps and gorillas with 3 lumbar vertebrae; either 2 or 3 is middle for columns with 4 lumbar vertebrae, so we decided to use L2 for consistency.

Lines 384-386: For the 3D analyses, did you also select exclusively those individuals with four lumbar vertebrae? Eastern gorillas have a vertebral formula of 7:13:3:6, that is, they usually have three lumbar vertebrae. Also, in Line 386, add graueri to Gorilla beringei, otherwise, that could also refer to mountain gorillas (Gorilla beringei beringei).

No, since middle lumbar vertebrae were being analyzed, we used L2 for all great apes. We realize this includes the antepenultimate lumbar vertebra of many western gorillas, chimps, and orangs, but the penultimate lumbar vertebra of most eastern gorillas (and some western gorillas and chimps), but we did not see evidence for separation between specimens with 3 and 4 lumbar vertebrae. Regarding eastern gorillas, we refer to the species *Gorilla beringei* and not specific subspecies. Our samples include both *G. beringei graueri* and *G. b. beringei*.

Line 387: Be specific with the anatomical determination of the vertebra, “U.W.88-233 is a complete third lumbar vertebra”.

“third” added.

Line 388: Lumbar block is not appropriated. It could be the block including the fossil lumbar vertebrae or something similar.

We have taken the reviewer’s advice and changed our references to both the “lumbar block” and the “lower thoracic block.”

409: I don't think a PCA is a good method to detect outliers. Your sample includes very diverse species, and an outlier may not be detected in such a huge variability. There are other, more specific, and appropriate methods to detect outliers. Please, modify this, and apply other functions such as plotOutliers from geomorph, or find.outliers from Morpho (or any other similar to these). The latter also includes a visual interface to detect possible mistakes in landmarking that are difficult to detect simply by plotting the sample looking for outliers.

We agree and have dropped this altogether.

412: Do you also use geomorph to perform the PCA? I am a bit confused with the plots you obtained. Also, I never saw those wireframes to visualize shape changes. Clarify where did you get them from. In addition, why don't you use those 3D models you have to obtain a more clear and visual representation of the shape's extreme morphologies? In the list of co-authors, some researchers usually represent shape variation showing modified 3D models.

Geomorph was used to perform the PCA. Wireframes were plotted using R. We prefer to use wireframes to display shape changes here for two reasons. First, wireframes display shape changes in *landmarks* directly, whereas the use of a 3D model involves interpolating the parts of the 3D model that were not explicitly quantified using landmarks. Second, the use of wireframes is also somewhat more practical since we would have to mirror the landmarks and the 3D template mesh to produce warped models. Finally, the authors you mention are not carrying out the 3D GM analyses in this case.

Line 412: Here I miss the goal of the regression analysis. This goes together with the lack of hypotheses to test, which could justify the methodological procedure of this work. Also, there is a very important lack of information about how this analysis was carried out. In the Results section, there is more information about the procedure than in the section “Methods” itself. Please, explain here step by step how did you perform this analysis. Also, in the Results section you mention that you grouped the species into genera pooling hominins together, but to perform a regression analysis including different species pooled into genera, previously you should remove the within-group variation.

The goal of the Procrustes distance-based analysis of variance (ANOVA) on GPA shape scores is to evaluate the effect of centroid size on middle lumbar vertebra shape (“allometry”). This goal fits into the larger purpose of the 3D GM analysis of the new *Australopithecus sediba* middle lumbar vertebra (U.W. 88-233) since shape differences between this fossil specimen and any extant taxon, such as *Homo sapiens*, could be explained by differences in size (“allometry”). Our results suggest that, while we detect a statistically significant effect of centroid size on middle lumbar vertebral shape within each group, the differences observed between *Au. sediba* and modern humans appears not to be explained by size alone. We have tried to explain this better. The model formula is: gpa cords ~ centroid size * genus. We have added a new sentence clarifying this and have attempted to better incorporate this portion of the analyses generally.

Line 415: In the same vein, you mention you carried out two analyses. But these are not analyses themselves. You used a different number of landmark coordinates to perform different analyses, which in addition you do not mention here. Please rewrite this paragraph and explain what you did.

We have reworded this part.

Line 595: Please describe the first “block” either from caudal to cranial or vice versa, but following an anatomical order. You start by the sacrum, then L5, then L4, L5 again, and finally S1. In the second block you follow the cranial-caudal direction for describing it, please, be consistent (same for the first block).

We have reorganized the caption to Figure 2.

Line 628: This is my concern about MH2 human-like vertebral body morphology. Once you remove the spinous process and the transverse processes, basically what we have is the vertebral body and the articular facets. The results indicate that australopiths are outside the modern humans' range of variation.

The reviewer makes a good point here and we have adjusted the results and discussion accordingly. Specifically, we state that our third hypothesis (about middle lumbar vertebra shape) is not fully supported due to both the costal process morphology and vertebral body shape.

Lines 637-642: All this information should be in a table. It is difficult to compare and follow all the information you show here.

We have created two new tables, which contain the extant comparative data (Table 2) and the fossil data (Table 3).

Line 645: Regarding the column with the values of gorillas and chimpanzees, clarify what Table are you referring to (Excel Table with the raw data for Figure 6 I guess).

We have modified the way we describe the combined wedging angle data.

Line 655: Thanks for providing all this information. This is just my point of view, and I will leave it in the hands of the AE, but I think a simple non-parametric analysis (e.g. Mann-Whitney u test) comparing these traits between extinct and extant species would have been very useful to assess specific significant differences (or similarities) between MH2 lumbar vertebrae and those of other australopiths, Neanderthals, modern humans, and great apes. The results could have (or not) demonstrate in specific traits (e.g. vertebral body) the similarities between modern humans and MH2 you propose as demonstrated by other methods (and I have my doubts after your results).

I understand why the reviewer would request such analyses, but they would be extraordinarily numerous – each of the 31 measurements would need to be compared in pairwise group comparisons (Mann-Whitney U test is a two sample test to my knowledge; Kruskall-Wallis would be the appropriate non-parametric test for multiple groups, but even still, 31 test would need to be run to include all measurements) – and more significantly, they would either need to be corrected for body mass (i.e., using estimated body mass or a geometric mean of some measurements, both of which are somewhat problematic) or would be largely arbitrary comparisons since of course MH2 and other female *Australopithecus* are smaller than gorillas, humans, and even chimpanzees in most measurements. Multivariate analyses that adjust data for size (i.e., Procrustes analysis in 3D GM, which we employ in our paper) are one way of circumventing the problematic nature of numerous univariate comparisons among groups characterized by size differences.

Line 856: Describe the views as in the other A, B, and C figures. Also, provide the number of the landmarks, if not all because of the lack of space, at least most of them.

We are not sure how the reviewer wants the figure caption reorganized, but think the description of the views is accurate. We also think the reviewer is asking for us to label the 48 landmarks (or at least most of them) in each of the five views (so adding upwards of 240 numbers). We tried but it becomes very full and we think distracting and impractical, so we have not added the landmark numbers. We think that interested readers will be able to match the landmark descriptions provided in Suppl Note 2 with the landmarks shown in what is now Figure 4. As with the choice to show wireframes rather than warped 3D models, we think this is a matter of preference. Thank you for the suggestion though.

Line 859: These analyses are not explained in the Methods section. Neither the obtained results. Include or remove them.

It was in fact referenced in both the Methods (penultimate paragraph) and in the Discussion (p. 10):

In the main text, we wrote, “We plotted standardized shape scores derived from a multivariate regression of shape on centroid size against centroid size to visualize shape changes (Drake and Klingenberg, 2008) (Supplementary Figure 4).” We have modified this sentence, which now says, “We plotted standardized shape scores derived from a multivariate regression of shape on centroid size against centroid size to visualize shape changes(24) using the RegScore method implemented in the ‘plotAllometry’ function in the ‘geomorph’ package (Supplementary Figure 4).”

This is simply one of a few ways to visualize shape allometries presented in the geomorph package.

Here is selected text from the geomorph pdf file:

“…describing a linear model (with procD.lm) that has an explicit definition of how shape allometries vary by group can be more informative. The following are the three most general models:

simple allometry: shape ~ size

common allometry, different means: shape ~ size + groups

unique allometries: shape ~ size * groups

[…] Either PredLine or RegScore can help elucidate divergence in allometry vectors among groups.”

Our analysis uses the “unique allometries” model formula (middle lumbar shape ~ centroid size * genus). The inclusion of the “interaction term” in the model formula above (* genus) enables us to evaluate size effects within each genus. In a simpler univariate context, including the interaction term specifies that we would estimate separate slopes *and* separate intercepts for each genus. The “common allometry, different means” model design would estimate a single slope (i.e., a “common allometry”) with different intercepts.

In total, the purpose and key takeaway of this analysis (Figure 7—figure supplement 1) is that the difference between U.W. 88-233 and the modern human sample cannot be explained by the smaller size of U.W. 88-233. We include this analysis to be as thorough as possible in the evaluation of morphometric affinities.

Line 860: “between U.W. 88-233 and extant and fossil middle lumbar vertebrae” Here only some of the extant species included in the sample are analyzed. Why? Specify which species you included.

We included these specific comparisons of the new *Au. sediba* specimen (U.W. 88-233) to *Homo sapiens* and *Pan troglodytes* for two reasons. First, the extant taxon to which *Au. sediba* is most closely related is *H. sapiens*, and obviously they are both bipeds, so that is a natural comparison, followed by *Pan troglodytes*. We could include *P. paniscus* as well, but our *P. troglodytes* sample size is larger, which is more ideal. We could pool *P. paniscus* and *P. troglodytes*, but that may falsely inflate the variance within *Pan* since the two groups differ slightly. Second, there is a practical issue of conducting pairwise comparisons of Procrustes distances across many groups.

Line 863: Could you please explain what is this within (Homo, Pan) (mean Procrustes distance?), or better said, the purpose of comparing that mean? Also, in this Figure 3 (B), why there is not lateral view as in Figure 3 (C)?

Supplementary Figure 3A shows the Procrustes distances between U.W. 88-233 and Sts 14, Shanidar 3, all modern humans sampled, and all chimpanzees sampled. It shows that U.W. 88-233 is most similar to Sts 14 in shape.

Supplementary Figure 3C shows pairwise comparisons of Procrustes distances for middle lumbar shape within our human and chimpanzee samples, and between our human and chimpanzee samples. The ‘within’ comparisons show the shape variation observed within humans and chimpanzees. The ‘between’ comparisons show the shape differences observed between our human and chimpanzee samples. This analysis shows that U.W. 88-233 has small shape differences from Sts 14 that are typically observed within humans or chimpanzees (i.e., they are not very different). In contrast, this analysis shows that U.W. 88-233 is much more distinct from Shanidar 3, with a magnitude of shape difference similar to that observed between humans and chimpanzees (i.e., they are very different).

This analysis provides a more in-depth comparison of the morphometric affinities of U.W. 88-233 that is a supplement to the main 3D GM analyses presented in the main text using PCA. We did not include a lateral view in Supplementary Figure 3B because the differences observed here were not very dramatic. In other words, we included a lateral view in 3D because, for example, the shape and orientation of the spinous and superior and inferior processes and vertebral body differ considerably between U.W. 88-233 and the average chimpanzee in lateral view (which comes as no surprise since one is a biped and the other is not).

Line 868: Here it would be interesting to comment that both australopiths plot within the chimpanzees' range of variation, and the Neandertal within (or in the limit) of modern humans.

The important thing to take away from Supplementary Figure 3 is that these are comparisons of *shape differences* represented by Procrustes distances. They are not comparisons of shape itself, which is only depicted in the 3D GM analysis in the main text. So, the line “UW 88-233-Sts14” in 3C is the Procrustes distance observed between U.W. 88-233 and Sts 14 (somewhere > 0.1), which falls within the ranges observed *within* humans and chimpanzees. The line “U.W. 88-233-Shanidar 3” in 3C is the Procrustes distance observed between U.W. 88-233 and Shanidar 3, which falls within the range of the Procrustes distances observed *between* humans and chimpanzees in our sample (i.e., the Au. sediba vertebra is as different from the Neandertal vertebra as is the typical human and chimpanzee; they are very different).

Reviewer #2 (Recommendations for the authors):Please always use the correct anatomical terms according to the current version of the Terminologia anatomica. The "lumbar transverse process" is an erroneous, antiquated and obsolete term that needs to be replaced by "costal process" throughout the paper. See also Strzelec, B., Chmielewski, P.P., Gworys, B., 2017. The Terminologia Anatomica matters: examples from didactic, scientific, and clinical practice. Folia Morphologica 76, 340-347.

We have read Strzelec et al. and consulted Terminologia Anatomica and understand the reviewer’s point here. Perhaps erroneously or inappropriately, we have published previously using “lumbar transverse process,” as have most researchers in biological anthropology and paleoanthropology. White et al. (2012) and Aiello and Dean (1990) similarly use “transverse process” in reference to this structure. However, we have now changed all cases to “costal process.” Due to the usage of transverse process in our field, we use “transverse” in parentheses in the first uses of costal process in our abstract and in the main text of the manuscript.

Likewise, the term "spinal canal" designates the canal that is formed by the vertebral foramina of several subsequent vertebrae for the spinal cord, but in the individual vertebra it would be termed "vertebral foramen".

We have changed “spinal canal” to “vertebral foramen” throughout.

A more stylistic issue is the use of the colloquial term "australopith". This word represents a sloppy mutilation of "pithekos", the Greek word for "ape". The syllable "pith" in australopith is nonsensical. Some people say they prefer this slang because they claim that the "australopithecine" would imply monophyly of the genera Australopithecus and Paranthropus. This is, however, not correct. Australopithecines is not only the English term for the former subfamily "Australopithecinae", but also for the subtribe "Australopithecina" of modern systematics. Therefore, I strongly urge the authors to replace "australopiths" with "australopithecines".

We would like to avoid a debate about colloquial terms in taxonomy, so we have replaced “australopith” with phrases like “members of the genus *Australopithecus*” and “early hominins included in this study.”

The authors compare the morphology of the newly described, complete third lumbar vertebra U.W. 88-233 of A. sediba to the "middle lumbar vertebrae" of modern humans, chimpanzees and gorillas. It is for me, however, unclear, what they exactly mean with "middle lumbar vertebra" because australopithecines and modern humans possess on average 5 lumbar vertebrae and chimpanzees and gorillas generally have (3-) 4. Could the authors therefore shortly discuss the issue of homology and how the outcome would be affected if U.W. 88-233 and the human L3 would be compared to L2 vs. L3 in the great apes.

The reviewer is correct that great apes with four lumbar vertebrae do not have a “middle” lumbar vertebra like those with three (L2) or humans with five lumbar vertebrae (L3). We chose to use the second lumbar vertebra to represent a hypothetical middle lumbar vertebra because it is “closer” to a human middle lumbar vertebra (L3) in that it is the penultimate lumbar vertebra. Additionally, for all great apes, L2 is used (vs. L2 in some and L3 in other individuals).

L213 "Humans and great apes separate along PC1, which is largely explained by vertebral body height": isn't it rather vertebral wedging?

The reviewer is correct that it is not just uniform height but also the added effect of wedging, so we have added wedging to the text.

L214 "U.W.88-233 and Sts 14 fall intermediately between modern humans and great apes along PC1": This should be changed to say that this applies to all australopithecines.

We have added “like other early hominins included in this study” to clarify this point.

L237 "a ventral curvature of the lumbar spine": a lordotic curve cannot be described as a ventral curvature-it is a ventrally convex curvature.

We have added “convex” here and elsewhere to make this correction. Thanks.

L241 "sum L2-L5 wedging" is awkward. Consider something like "total L2-L5 wedging angle".

We have changed this to “combined L2-5 wedging” and “combined wedging” throughout.

L248 "see Figure 6 caption": Unfortunately, I don't see the data for chimpanzees in in Figure 6., and also the caption to Figure 6 does not provide more relevant information. On the other hand, I don't think it is a good option to use figure captions to present additional information that is otherwise not contained within the manuscript.

Leaving chimpanzees out of the caption was an oversight – thank you for catching it. Given this reviewer’s requests and those of the other reviewer, we have converted these statistics to table form. Table 2 now shows all of the summary statistics so that the reader can appreciate the high variation in wedging angles within groups (human males and females and chimpanzees). Regarding Figure 6 itself, we only include chimpanzees in the combined wedging angle column. There are two reasons for this: (1) the figure shows L2-L5, whereas the elements represented by chimpanzees are necessarily L1-L4 (they do not have L5, and we excluded chimpanzees with just three lumbar vertebrae); and (2) if we include chimpanzee 95% confidence intervals and/or other elements (full range of variation for extant taxa), the figure becomes very busy and not easily readable. We prefer it this way and make it very clear that what are presented in the figure are confidence intervals, not the full range of variation. Additionally, Table 3 (formerly the figure caption) includes the standard deviation and range of the data – for both humans and chimpanzees at individual and combined levels – and our Source Data files contain all of the wedging angle data for individual specimens, so the reader can examine these.

L249 "Patterns of change across lumbar levels demonstrate that MH2's vertebrae transition from ventral (kyphotic) wedging at the L2 (most similar to male modern humans) and L3 (most similar to female modern humans) levels to dorsal (lordotic) wedging at L4. At the L4 level MH2 is most similar to female modern humans and female…": In my opinion, inferences for a greater similarity of the vertebral wedging patterns of fossil hominins to a specific modern humans sex is overinterpreting the data. Thus, the considerable variability of these wedging patterns within modern humans needs to be taken into account. In fact, the data presented in Figure 6 suggest that both males and females transition from ventral to dorsal wedging between L3 and L4. This also relates to the quite misleading use of the 95% confidence intervals of the means in Figure 6 rather than the 95% confidence intervals of the sample data (see my comment to Figure 6).

The reviewer’s point is well taken. In fact, males and females are not significantly different at the L3-5 levels, which can be appreciated by the overlap of 95% confidence intervals of the mean. We have left the 95% CIs so readers can appreciate this, but we have also added in 2 standard deviations of the mean. At L2 and in combined wedging, human males and females are significantly different, apparent from non-overlapping 95% confidence intervals. We strongly prefer to show 95% CIs of the mean along with 2 standard deviations of the mean for that reason. We also report the standard deviation and range of all the data in Table 2 so that readers understand that the data are characterized by a high degree of variation. NOTE: we caught a mistake in the original figure and corrected it in the revision. Although our data in Source Data were correct, we used incorrect data for the combined wedging values for Neandertals. Although their positions do not change much, they are different now. We regret the oversight and wanted to bring it to your attention. It was in revisiting this figure that we noticed the mistake, so thank you.

L270 Maybe the authors can briefly discuss the functional significance of the pyramidal configuration for lumbar lordosis and cite the corresponding literature. It would also be good to discuss Oreopithecus in this context, for which a similar pyramidal configuration has been reported (Köhler, M., Moyà-Solà, S., 1997. Ape-like or hominid-like – the positional behavior of Oreopithecus bambolii reconsidered. Proceedings of the National Academy of Sciences of the United States of America 94, 11747-11750.)

We have added this reference along with Russo and Shapiro’s (2013) refutation of it.

L288 "ample surface area": isn't the length of the lever arm of the costal processes more relevant than the relatively moderate increase in surface area (if at all) compared to that of modern humans? And why do australopithecines need more powerful trunk musculature than modern humans? I also don't understand, why a cranial orientation of the costal processes should increase the "moment arms and torque generation capabilities of psoas major and quadratus lumborum". Isn't the lever arm only related to its length perpendicular to the force vector? I therefore think that the cranial orientation of the costal process must have another explanation.

The question of why *A. sediba* would “need” more powerful trunk musculature than modern humans is an interesting one, but the biomechanical effect of elongated and differently oriented costal processes in *A. sediba* seems clear. The overall morphology of the *A. sedib*a costal processes is consistent with increased moment arms for both psoas major and quadratus lumborum; this includes their length and more cranial and ventral orientations. It’s not the *cranial* orientation per se that creates a longer moment arm, but in our view, this trait is probably not morphologically independent of other costal process traits. Furthermore, we don’t consider this interpretation to be a complete explanation for the expression of this trait, and there could be additional, non-mutually exclusive explanations. We have provided a new supplementary figure (Figure 7—figure supplement 2) that depicts hypothetical muscle force vectors and moment arms. We state in the main text that more work needs to be done in this area.

L290 There is no "middle lumbar fascia", but the anterior and middle layers of the thoracolumbar fascia insert on the costal processes. On the other hand, the m. obliquus externus abdominis has generally no contact to the thoracolumbar fascia in modern humans as far as I know.

We have made this correction. Thank you.

L295 Please rephrase. Muscles cannot "support" the pelvis. Support means "bear all or part of the weight of something", but as muscles cannot counteract compressive forces they cannot bear weight. Moreover, the support of a structure must be below the structure itself.

We have rephrased this part of the sentence.

L299 Please also see Tardieu et al. (2017. How the pelvis and vertebral column became a functional unit in human evolution during the transition from occasional to permanent bipedalism? Anatomical Record 300, 912-931) who show that the pelvic incidence of alternative pelvic reconstructions of MH2 falls well within the range of modern humans.

We have incorporated this reference into our discussion and regret missing it originally.

L303 The analysis of La Chapelle-aux-Saints demonstrated that Neanderthals possessed a well-developed lordosis similar to modern humans, see Haeusler et al. (2019. Morphology, pathology and the vertebral posture of the La Chapelle-aux-Saints Neandertal. Proceedings of the National Academy of Sciences of the United States of America 116, 4923-4927). Moreover, the reported low pelvic incidence of SH1 is probably misleading because it is due to a lumbosacral transitional anomaly, see Haeusler, M., 2019. Spinal pathologies in fossil hominins, in: Been et al.. (Eds.), Spinal Evolution: morphology, function, and pathology of the spine in hominoid evolution. Springer, Cham, pp. 213-245.

We do include a “but see” and cite Haeusler et al. (2019), and we are sympathetic to this perspective; however, it is currently not the consensus view.

L308 Please see my other comments. I don't think it is possible to differentiate the sex based on the wedging values.

Two sets of co-authors on this manuscript have published manuscripts showing significant differences between human sexes in lumbar wedging (Ostrofsky and Churchill, 2015; García-Martínez et al., 2020). We also demonstrate significant differences here, at least at the L2 level and combined L2-L5 levels. Another subset of authors, including the first author, has recently completed a study on lumbar wedging and inferior articular facet angles and finds strong evidence for sex differences. Therefore, we acknowledge that vertebral wedging, in particular, are characterized by high variation within sex, but this variation does not swamp the differences between sexes, which are frequently significant.

L309 This is not true. Schiess et al. (2014. Revisiting scoliosis in the KNM-WT 15000 Homo erectus skeleton. Journal of Human Evolution 67, 48-59.) demonstrated that KNM-WT 15000 (Homo erectus) possesses an even slightly stronger dorsally wedged L5 vertebra than MH2. Nevertheless, it was still within the 95% range of variation of modern humans (see their Table 4 and Figure 5D).

Solved by adding “adult”.

L337 Please mention also that its presence has been claimed for Oreopithecus (see above).

We now discuss Oreopithecus in this area.

L354 „high-quality" seems to miss "scans" or something similar. It would be useful if the authors could indicate where they come from. At least the scans of Shanidar3 that are distributed by the Smithsonian can nowadays hardly be called "high quality". Please also note that the Shanidar vertebrae are extensively reconstructed.

Address from R1. High-quality casts from E. Trinkaus.

L357 I'm not aware that any of the fossils used in the present study are curated at the NHM London. Please check.

Correct, thanks. Likewise for Musée de l’Homme. Both removed.

L365 I wasn't aware that also modern human skeletons are curated at the Musée Royale de l'Afrique Centrale. Please check.

Right again. Removed. These were errors and we appreciate the reviewer bringing them to our attention.

L368 Please note that this is a basic trigonometry approach that has been used by DiGiovanni et al., and I therefore recommend to say this rather than citing DiGiovanni et al. There is also no fancy "wedging angle equation" necessary, but the simple use of the tangent. On the other hand, this equation is not provided in Table 1 as the text suggests, so it might be good to rephrase this sentence.

Understood, and we have reworded this part while retaining the citation. The “equation” was included in Supplementary Note 1 (now the Appendix) as Linear and Angular Measurements #7.

L372 Why this? What is needed here is the 95% range of variation or more precisely, the 3rd to 97th inter-percentile range.

As stated in more detail below, we now show 2 standard deviations of the mean, which encapsulates 95% of the data by definition. We do not show these in the combined column because they are very large and would require expanding the size of the y-axis significantly, but we do make clear that (1) what are shown are 95% confidence intervals of the mean and (2) we provide the standard deviation and range for human males and females and for chimpanzees in Table 3. In addition, the data for each individual are provided in the Source Data file.

Figure 6 Please show the 95% confidence intervals of the sample data, not the 95% confidence intervals of the means for modern humans males and females. This misleading and does not acknowledge the considerable range variation that is typical for these wedging data (see e.g., Figure 5D in Schiess et al. 2014).

As stated previously, we have added 2 standard deviations of the mean to each sex at each vertebral level (L2-L5). We did not include the 95% confidence intervals of the means to be misleading, but rather to make the figure more readable. Previously, we had included the full range of variation (as boxplots essentially), but the figure looked very crowded. We think we have found a way to present 95% of the data and still keep it readable. We do think it is interesting that two female Australopithecus fall within the 95% CIs of the female human mean, whereas one male Australopithecus falls within that of the male human mean. However, we realize that larger samples are required to say anything definitive about sexual dimorphism in fossil hominin lumbar wedging.

L637-642 Please consider to report these values in a table rather than in the figure caption.

We now include the extant data in Table 2 and the fossil data in Table 3.

Figure 7 Please use the correct scientific abbreviations! On the other hand I don't think it is still fine to use H. neanderthalensis for the Neanderthals since genetic data clearly demonstrate interbreeding with modern humans.

We have corrected the abbreviations and refer to “Neandertals” and “Modern humans” rather than *H. neanderthalensis* and *H. sapiens*, although we do retain the latter in the main text to refer to modern humans only.

[Editors' note: further revisions were suggested prior to acceptance, as described below.]

Reviewer #1 (Recommendations for the authors):I am satisfied that the authors have addressed most of my concerns, especially those regarding the interpretation of their results. I am also glad about the inclusion of the hypotheses, which I think help to better understand and follow the purpose of this work. The authors have explained in greater detail some methodological aspects that needed some clarification. All in this, the discussion is much more solid and coherent with the results than the previous version. In general, the authors have done a great good job. As I assessed in my first revision, these fossils are a great contribution to paleoanthropology, especially to the study of the evolution of the vertebral column in hominins. Thus, I recommend the publication of this manuscript after correcting a few details.First, I have some doubts about one aspect of the rebuttal document:The only point I think was not clarified in my previous revision referred to Supplementary Figure 3 (from the first version). I wrote: "Line 859: These analyses are not explained in the Methods section. Neither the obtained results. Include or remove them.".The authors have clarified that: "It was in fact referenced in both the Methods (penultimate paragraph) and in the Discussion (p. 10):. In the main text, we wrote, "We plotted standardized shape scores derived from a multivariate regression of shape on centroid size against centroid size to visualize shape changes (Drake and Klingenberg, 2008) (Supplementary Figure 4)."But this refers to Supplementary figure 4 (Line 868), and not to Supplementary figure 3, which corresponds with Line 859. I appreciate they clarified this part, but the analyses and results from this Supplementary Figure 3 are still not explained in the manuscript. This figure reads "Procrustes distances and mean differences…" but neither a reference to Procrustes distances nor to mean differences appear in the entire manuscript apart from this footnote. In the rebuttal to my concern about Line 863, which also refers to this figure, they explain in detail what this figure means, and I appreciate it, but this should also be in the manuscript.

We apologize for addressing the wrong issue in our last revision. We fully understand what the reviewer is asking for and have added a paragraph to the Results section explaining what is now Figure 7—figure supplement 1. Thank you for taking the time to bring this issue to our attention again.

Reviewer #2 (Recommendations for the authors):The revised manuscript has improved in many aspects. Particularly, it is now more than a simple exploratory study, having a greater focus on hypothesis testing. However, the wrong use of statistics in the analysis and hypothesis testing of the wedging angles still represents a major issue that needs to be addressed.Thus, the first hypothesis (H1a) of the current study is that there is no difference between the wedging angles of MH2 and modern humans, and another hypothesis, which I call H1b, is that they are distinct from extant great apes. To test hypothesis H1a, the authors compare the wedging angles of the MH2 lumbar vertebrae with the 95% confidence intervals of the mean female and the mean male modern human wedging angles. Generally, the 95% confidence intervals about the means are useful if the purpose is to compare two samples and to explore whether their means are different. However, the distribution of the wedging angles and thus their mean values are unknown for A. sediba (or any other fossil hominin species). Only those of a single A. sediba individual, MH2, are known, and a comparison of the means of A. sediba with those of modern humans is therefore not possible. We can only compare the MH2 specimen with the modern human sample by hypothesizing that there is no difference between A. sediba and *H. sapiens* for this trait (=H1a). Under this assumption, the data point for A.sediba would therefore fall with a 95% probability within a certain range. This is represented by the 95% confidence interval of the sample (also known as the 95% prediction interval). Thus, the 95% prediction interval represents the range of values that likely contains the value of a new observation given the distribution of the comparative sample. This 95% prediction interval can be approximated by 2 standard deviations (more precisely, it would be 1.96×standard deviations), and the authors now also show this range in their Figure 5, but unfortunately they don't use this interval further and don't discuss it within the text.In fact, Figure 5 shows that the wedging angles of all lumbar vertebrae of MH2 fall within the 95% prediction intervals of both modern human males and females. The same is true for all other analysed fossil hominins, except for Shanidar 3 and Kebara 2, whose wedging angles of L2 fall only within the male range of the current sample. Because we don't know the 95% confidence intervals about the means of the A. sediba wedging angles (or those of A. africanus, etc.), it is irrelevant whether MH2 (or Sts 14 or StW 431) lies closer to the female or the male means of modern humans for some vertebrae, as they are only some individuals. The corresponding sections in the text (L316-319 and L350-366) should therefore be rephrased accordingly. Likewise, the right side of Figure 5 needs to be adapted to show the 95% prediction intervals rather the 95% confidence intervals of the means.Hypothesis H1b (that the wedging angles of MH2 are distinct from extant great apes) is only marginally addressed as far as I can see. Thus, wedging angles are only reported in Table 2 for chimpanzees (and thus only for one of three great ape genera). Nevertheless, it seems that the wedging angles of vertebrae L2-L4 of MH2 are well within the 95% prediction intervals for chimpanzees (as approximated by the means {plus minus} 2 SD). Does this therefore mean that lumbar lordosis of MH2 or other fossil hominins cannot statistically be differentiated from that of chimpanzees? Can the authors expand on this? It also would be helpful if the means and the 95% prediction intervals for chimpanzees (and if possible gorillas and orangutans) are included in figure 5 (or in an additional figure).

We have extensively revised this figure and removed the 95% CIs. To accommodate great apes and 95% PIs for all extant groups, we split Figure 5 into a main figure and two supplementary figures.

Regarding my suggestion to include KNM-WT 15000 into the study, I agree with the authors that this is not so easy for the 3D GM analyses due to its subadult age. However, I still maintain that the addition of KNM-WT 15000 would be fundamental to the interpretation of the wedging angles as it shows that the strong lordotic wedging of L5 is not exceptional in MH2 and Kebara 2 (see Schiess et al. 2014). The subadult age of KNM-WT 15000 explains of course the missing vertebral ring apophyses, but this does not affect the wedging angles of the vertebral bodies since the ring apophyses are flat.

We have added KNM-WT 15000 (the “Nariokotome Boy”) to the wedging angle figures. Please note that we have also corrected data points in the figures and in Table 3 that previously represented errors, the most major of which was our source data on combined *Pan* and *Gorilla* wedging values. We had previously inadvertently used only the last three lumbar vertebrae, whereas now we present – in the Figure 5 and its supplements and in Table 2 and its source data file – the combined value for all four lumbar vertebrae.